# A Bandit Learning Algorithm and Applications to Auction Design

**Nguyễn Kim Thắng**
IBISC, Univ. Evry, University Paris-Saclay, France
kimthang.nguyen@univ-evry.fr

## Abstract

We consider online bandit learning in which at every time step, an algorithm has to make a decision and then observe only its reward. The goal is to design efficient (polynomial-time) algorithms that achieve a total reward approximately close to that of the best fixed decision in hindsight. In this paper, we introduce a new notion of $(\lambda, \mu)$-concave functions and present a bandit learning algorithm that achieves a performance guarantee which is characterized as a function of the concavity parameters $\lambda$ and $\mu$. The algorithm is based on the mirror descent algorithm in which the update directions follow the gradient of the multilinear extensions of the reward functions. The regret bound induced by our algorithm is $\widetilde{O}(\sqrt{T})$ which is nearly optimal.

We apply our algorithm to auction design, specifically to welfare maximization, revenue maximization, and no-envy learning in auctions. In welfare maximization, we show that a version of fictitious play in smooth auctions guarantees a competitive regret bound which is determined by the smooth parameters. In revenue maximization, we consider the simultaneous second-price auctions with reserve prices in multi-parameter environments. We give a bandit algorithm which achieves the total revenue at least $1/2$ times that of the best fixed reserve prices in hindsight. In no-envy learning, we study the bandit item selection problem where the player valuation is submodular and provide an efficient $1/2$-approximation no-envy algorithm.

## 1 Introduction

In online learning, the goal is to design algorithms which are robust in dynamically evolving environments by applying optimization methods that learn from experience and observations. Characterizing conditions, or in general discovering regularity properties, under which efficient online learning algorithms with performance guarantee exist is a major research agenda in online learning. In this paper, we consider this line of research and present a new regularity condition for the design of efficient online learning algorithms. Subsequently, we establish the applicability of our approach in auction design.

**General online problem.**   At each time step $t = 1, 2, \ldots$, an algorithm chooses $\boldsymbol{x}^t \in [0,1]^n$. After the algorithm has committed to its choice, an adversary selects a function $f^t : [0,1]^n \to [0,1]$ that subsequently induces the reward of $f^t(\boldsymbol{x}^t)$ for the algorithm. In the problem, we are interested in the bandit setting that at every time $t$, the algorithm observes only its reward $f^t(\boldsymbol{x}^t)$. The goal is to efficiently achieve the total gain approximately close to that obtained by the best decision in hindsight.

We consider the following notion of regret which measures the performance of algorithms. An algorithm is $(r, R(T))$-*regret* if for arbitrary total number of time steps $T$ and for any sequence of reward functions $f^1, \ldots, f^T \in \mathcal{F}$, $\sum_{t=1}^{T} f^t(\boldsymbol{x}^t) \geq r \cdot \max_{\boldsymbol{x} \in [0,1]^n} \sum_{t=1}^{T} f^t(\boldsymbol{x}) - R(T)$. We also

say that the algorithm achieves a $r$-regret bound of $R(T)$. In general, one seeks algorithms with $(r, R(T))$-regret such that $r > 0$ is as large as possible (ideally, close to 1) and $R(T)$ is sublinear as a function of $T$, i.e., $R(T) = o(T)$. We also call $r$ as the *approximation ratio* of the algorithm.

We introduce a regularity notion that generalizes the notion of concavity. The new notion, while simple, is crucial in our framework in order to design efficient online learning algorithms with performance guarantee.

**Definition 1** *A function $F$ is $(\lambda, \mu)$-concave if for every vectors $\boldsymbol{x}$ and $\boldsymbol{x}^*$,*

$$\langle \nabla F(\boldsymbol{x}), \boldsymbol{x}^* - \boldsymbol{x} \rangle \geq \lambda F(\boldsymbol{x}^*) - \mu F(\boldsymbol{x}) \tag{1}$$

Note that a concave function is $(1, 1)$-concave. A non-trivial example, shown in the paper, is the $(1, 2)$-concavity of the multilinear extension of a monotone submodular function.

## 1.1 The Main Algorithm

We aim to design a bandit algorithm for the general online problem with emphasis on auctions. Bandit algorithms have been widely studied in online convex optimization [17] but in the context of auction design, standard approaches have various limits. The main issues are: (1) the non-concavity of the (reward) functions, and (2) the intrinsic nature of the bandit setting (only the value $f^t(\boldsymbol{x}^t)$ is observed). We overcome these issues by the approach which consists of lifting the search space (of the solutions) and the reward functions to a higher dimension space and considering the multilinear extensions of the reward functions in that space. Concretely, we consider a sufficiently dense lattice $\mathcal{L}$ in $[0, 1]^n$ such that every point in $[0, 1]^n$ can be approximated by a point in $\mathcal{L}$. Then, we lift all lattice points in $\mathcal{L}$ to vertices of a hypercube in a high dimension space. Subsequently, we consider the multilinear extensions of reward functions $f^t$ in that space. This procedure enables several useful properties, in particular the $(\cdot, \cdot)$-concavity, that hold for the multilinear extensions but not necessarily for the original reward functions. (For example, the multilinear extension of a monotone submodular function is always $(1, 2)$-concave but the submodular function is not.) The introduction of $(\cdot, \cdot)$-concavity and the use of multilinear extensions constitute the novel points in our approach compared to the previous ones. This allows us to bound the regret of our algorithm which is based on the classic mirror descent with respect to the gradients of the multilinear extensions.

**Informal Theorem 1** *Let $f^t : [0, 1]^n \rightarrow [0, 1]$ be the reward function at time $1 \leq t \leq T$ and let $F^t$ be the multilinear extension of the discretization of $f^t$ on a lattice $\mathcal{L}$. Assume that $f^t$'s are L-Lipschitz and $F^t$'s are $(\lambda, \mu)$-concave. Then, there exists a bandit algorithm achieving*

$$\sum_{t=1}^{T} \mathbb{E}\big[f^t(\boldsymbol{x}^t)\big] \geq \frac{\lambda}{\mu} \cdot \max_{\boldsymbol{x} \in [0,1]^n} \sum_{t=1}^{T} f^t(\boldsymbol{x}) - O\big(\max\{\lambda/\mu, 1\} Ln^{3/2}(\log T)^{3/2}(\log \log T)\sqrt{T}\big).$$

The formal statement corresponding to the above informal theorem is Theorem 2. By this theorem, determining the performance guarantee is reduced to computing the concave parameters. Moreover, the regret bound of $\widetilde{O}(\sqrt{T})$ is nearly optimal that has been proved in the context of online convex optimization (for concave functions, i.e., $(1, 1)$-concave functions). The approach is convenient to derive bandit learning algorithms in the context of auction design as shown in the applications.

## 1.2 Applications to Auction Design

In a general auction design setting, each player $i$ has a *valuation* (or *type*) $v_i$ and a set of actions $\mathcal{A}_i$ for $1 \leq i \leq n$. Given an action profile $\boldsymbol{a} = (a_1, \ldots, a_n)$ consisting of actions chosen by players, the auctioneer decides an allocation $\boldsymbol{o}(\boldsymbol{a})$ and a payment $p_i(\boldsymbol{o}(\boldsymbol{a}))$ for each player $i$. Note that for a fixed auction $\boldsymbol{o}$, the outcome $\boldsymbol{o}(\boldsymbol{a})$ of the game is completely determined by the action profile $\boldsymbol{a}$. Then, the *utility* of player $i$ with valuation $v_i$, following the quasi-linear utility model, is defined as $u_i(\boldsymbol{o}(\boldsymbol{a}); v_i) = v_i(\boldsymbol{o}(\boldsymbol{a})) - p_i(\boldsymbol{o}(\boldsymbol{a}))$. The *social welfare* of an auction is defined as the total utility of all participants (the players and the auctioneer): $\text{Sw}(\boldsymbol{o}(\boldsymbol{a}); \boldsymbol{v}) = \sum_{i=1}^{n} u_i(\boldsymbol{o}(\boldsymbol{a}); v_i) + \sum_{i=1}^{n} p_i(\boldsymbol{a})$. The total revenue of the auction is $\text{Rev}(\boldsymbol{o}(\boldsymbol{a}), \boldsymbol{v}) = \sum_{i=1}^{n} p_i(\boldsymbol{o}(\boldsymbol{a}))$. When $\boldsymbol{o}$ is clear in the context, we simply write $v_i(\boldsymbol{a}), u_i(\boldsymbol{a}; v_i), p_i(\boldsymbol{a}), \text{Sw}(\boldsymbol{a}; \boldsymbol{v}), \text{Rev}(\boldsymbol{a}, \boldsymbol{v})$ instead of $v_i(\boldsymbol{o}(\boldsymbol{a})), u_i(\boldsymbol{o}(\boldsymbol{a}); v_i), p_i(\boldsymbol{o}(\boldsymbol{a})), \text{Sw}(\boldsymbol{o}(\boldsymbol{a}); \boldsymbol{v}), \text{Rev}(\boldsymbol{o}(\boldsymbol{a}), \boldsymbol{v})$, respectively. In the paper, we consider two standard objectives: welfare maximization and revenue maximization. Note that in revenue maximization, we call players as bidders.

### 1.2.1 Fictitious Play in Smooth Auctions

We consider adaptive dynamics in auctions. In the setting, there is an underlying auction $\boldsymbol{o}$ and there are $n$ players, each player $i$ has a set of actions $\mathcal{A}_i$ and a valuation function $v_i$ taking values in $[0, 1]$ (by normalization). In every time step $1 \leq t \leq T$, each player $i$ selects a strategy which is a distribution in $\Delta(\mathcal{A}_i)$ according to some given adaptive dynamic. After all players have committed their strategies, which result in a strategy profile $\boldsymbol{\sigma}^t \in \Delta(\mathcal{A})$, the auction induces a social welfare $\mathrm{Sw}(\boldsymbol{o}, \boldsymbol{\sigma}^t) := \mathbb{E}_{\boldsymbol{a} \sim \boldsymbol{\sigma}^t}\big[\mathrm{Sw}(\boldsymbol{o}(\boldsymbol{a}); \boldsymbol{v})\big]$. In this setting, we study the total welfare achieved by the given adaptive dynamic comparing to the optimal welfare. This problem can be cast in the online optimization framework in which at time step $t$, the player strategy profile corresponds to the decision of the algorithm and subsequently, the gain of the algorithm is the social welfare induced by the auction w.r.t the strategy profile.

Smooth auctions is an important class of auctions in welfare maximization. The smoothness notion has been introduced [32, 28] in order to characterize the efficiency of (Bayes-Nash) equilibria of auctions. It has been shown that several auctions in widely studied settings are smooth; and many proof techniques analyzing equilibrium efficiency can be reduced to the smooth argument.

**Definition 2 ([32, 28])** *For parameters $\lambda, \mu \geq 0$, an auction is $(\lambda, \mu)$-smooth if for every valuation profile $\boldsymbol{v} = (v_1, \ldots, v_n)$, there exist action distributions $\overline{D}_1(\boldsymbol{v}), \ldots, \overline{D}_n(\boldsymbol{v})$ over $\mathcal{A}_1, \ldots, \mathcal{A}_n$ such that, for every action profile $\boldsymbol{a}$, $\sum_{i=1}^{n} \mathbb{E}_{\overline{a}_i \sim \overline{D}_i(\boldsymbol{v})}\big[u_i(\overline{a}_i, \boldsymbol{a}_{-i}; v_i)\big] \geq \lambda \cdot \mathrm{Sw}(\overline{\boldsymbol{a}}; \boldsymbol{v}) - \mu \cdot \mathrm{Sw}(\boldsymbol{a}; \boldsymbol{v})$ where $\boldsymbol{a}_{-i}$ is the action profile similar to $\boldsymbol{a}$ without player $i$.*

It has been proved that if an auction is $(\lambda, \mu)$-smooth then every Bayes-Nash equilibrium of the auction has expected welfare at least $\lambda/\mu$ fraction of the optimal auction [28, 32]. Moreover, the smoothness framework does extend to individually-vanishing-regret dynamics. A sequence of actions profiles $\boldsymbol{a}^1, \boldsymbol{a}^2, \ldots$, is an *individually-vanishing-regret sequence* if for every player $i$ and action $a_i'$, $\lim_{T \to \infty} \frac{1}{T} \sum_{t=1}^{T} \big[u_i(a_i', \boldsymbol{a}_{-i}^t; v_i) - u_i(\boldsymbol{a}^t; v_i)\big] \leq 0$.

However, several interesting dynamics are not guaranteed to have the individually-vanishing-regret property. In a recent survey, Roughgarden et al. [30] have raised a question whether adaptive dynamics without the individually-vanishing-regret condition can achieve approximate optimal welfare. Among others, *fictitious play* [5] is an interesting, widely-studied dynamic which attracts a significant attention in the community.

In the paper, we consider a version of fictitious play in smooth auctions, namely Perturbed Discrete Time Fictitious Play (PDTFP). In general, it is not known whether this dynamic has individually-vanishing-regret. Despite that fact, using our framework, we prove that given an offline $(\lambda, \mu)$-smooth auction, PDTFP dynamic achieves a $\lambda/(1 + \mu)$ fraction of the optimal welfare.

**Informal Theorem 2** *If the underlying auction $\boldsymbol{o}$ is a $(\lambda, \mu)$-smooth then the PDTFP dynamic achieves $\left(\frac{\lambda}{1+\mu}, R(T)\right)$-regret where $R(T) = O\big(\frac{\sqrt{T}}{1+\mu}\big)$.*

### 1.2.2 Revenue maximization in Multi-Dimensional Environments

We consider online simultaneous second-price auctions with reserve prices in *multi-dimensional* environments. In the setting, there are $n$ bidders and $m$ items to be sold to these bidders. At every time step $t = 1, 2, \ldots, T$, the auctioneer selects reserve prices $r_i^t = (r_{i1}^t, \ldots, r_{im}^t)$ for each bidder $i$ where $r_{ij}^t$ is the reserve price of item $j$ for bidder $i$. Each bidder $i$ for $1 \leq i \leq n$ has a (private) valuation $v_i^t : 2^{[m]} \to \mathbb{R}^+$ over subsets of items. After the reserve prices have been chosen, every bidder $i$ picks a bid vector $b_i^t$ where $b_{ij}^t$ is the bid of bidder $i$ on item $j$ for $1 \leq j \leq m$. Then the auction for each item $1 \leq j \leq m$ works as follows: (1) remove all bidders $i$ with $b_{ij}^t < r_{ij}^t$; (2) run the second price auction on the remaining bidders to determine the winner of item $j$ — the bidder with highest non-removed bid on item $j$; and (3) charge the winner of item $j$ the price which is the maximum of $r_{ij}^t$ and the second highest bid among non-removed bids $b_{ij}^t$. The objective of the auctioneer is to achieve the total revenue approximately close to that achieved by the best fixed reserve-price auction. Note that in the bandit setting, the auction is given as a blackbox and at every time step, the auctioneer observes only the total revenue (total price) without knowing neither the bids of bidders nor the winner/price of each item. The setting enhances, among others, the privacy of bidders.

The second-price auctions with reserve prices in *single-parameter* environments have been considered by Roughgarden and Wang [29] in full-information online learning. Using the Follow-the-Perturbed-Leader strategy, they gave a polynomial-time online algorithm that achieves half the revenue of the best fixed reserve-price auction minus a term $O(\sqrt{T}\log T)$ (so their algorithm is $(1/2, O(\sqrt{T}\log T))$-regret in our terminology). The problem we consider cannot be reduced to applying their algorithm on $m$ separated items since (1) bids on different items might be highly correlated (due to bidders' valuations); and (2) in the bandit setting for multiple items, the auctioneer know only the total revenue (not the revenue from each item). Using our framework, we prove the following result.

**Informal Theorem 3** *There exist a bandit algorithm that achieves $\left(1/2, O(m\sqrt{nmT}\log T)\right)$-regret for revenue maximization in multi-parameter environments.*

### 1.2.3 Bandit No-Envy Learning in Auctions

The concept of *no-envy learning* in auctions has been introduced by Daskalakis and Syrgkanis [10] in order to maintain approximate welfare optimality while guaranteeing computational tractability. The concept is inspired by the notion of *Walrasian equilibrium*. Intuitively, an allocation of items to buyers together with a price on each item forms a Walrasian equilibrium if no buyer envies other allocation given the current prices. In the paper, we consider no-envy bandit learning algorithms for the following *online item selection* problem.

In the problem, there are $m$ items and a player with monotone valuation $v : 2^{[m]} \to \mathbb{R}^+$. At every time step $1 \le t \le T$, the player chooses a subset of items $S^t \subset [m]$ and the adversary picks adaptively (probably depending on the history up to time $t-1$ but not on the current set $S^t$) a threshold vector $\boldsymbol{p}^t$. The player observed the total price $\sum_{j \in S^t} p_j^t$ and gets the reward of $v(S^t) - \sum_{j \in S^t} p_j^t$. A learning algorithm for the online item selection problem is a *r-approximate no-envy* [10] if for any adaptively chosen sequence of threshold vectors $\boldsymbol{p}^t$ for $1 \le t \le T$, the sets $S^t$ for $1 \le t \le T$ chosen by the algorithm satisfy $\mathbb{E}\left[\sum_{t=1}^T \left(v(S^t) - \sum_{j \in S^t} p_j^t\right)\right] \ge \max_{S \subseteq [m]} \sum_{t=1}^T \left(r \cdot v(S) - \sum_{j \in S} p_j^t\right) - R(T)$ where the regret $R(T) = o(T)$.

Daskalakis and Syrgkanis [10] considered the problem in the full-information setting (i.e., at every time step $t$, the player observes the whole vector $\boldsymbol{p}^t$) where the valuation $v$ is a coverage function[1] and gave an $(1 - 1/e)$-approximate no-envy algorithm with regret bound $O(\sqrt{T})$. The algorithm is designed via the convex rounding scheme [12], a technique which has been used in approximation algorithms and in truthful mechanism design. In this paper, we consider *submodular* valuations, a more general and widely-studied class of valuations. A valuation $v : 2^{[m]} \to \mathbb{R}^+$ is *submodular* if for any sets $S \subset T \subset [m]$, and for every item $j$, it holds that $v(S \cup j) - v(S) \ge v(T \cup j) - v(T)$. Using our framework, we derive the following result.

**Informal Theorem 4** *There exist an $\left(1/2, O(m^{3/2}\sqrt{T}\log(mT))\right)$-regret no-envy learning algorithm for the bandit item selection problem where the player valuation is submodular.*

### 1.3 Related Work

There is large literature on online learning and auction design. In this section, we summarize and discuss only works directly related to ours. The interested reader can refer to [31, 17] for online learning and to [30] (and references therein) for auction design.

**Online/Bandit Learning.** Online learning, or online convex optimization, is an active research domain. The first no-regret algorithm was given by Hannan [16]. Subsequently, Littlestone and Warmuth [23] and Freund and Schapire [14] gave improved algorithms with regret $\sqrt{\log(|\mathcal{A}|)}o(T)$ where $|\mathcal{A}|$ is the size of the action space. Kalai and Vempala [20] presented the first efficient online algorithm, called *Follow-the-Perturbed-Leader* (FTPL), for linear objective functions. The strategy consists of adding perturbation to the cumulative gain (payoff) of each action and then selecting the action with the highest perturbed gain. This strategy has been generalized and successfully applied to several settings [18, 33, 10, 11]. Specifically, FTPL and its generalized versions have been used to

design efficient online no-regret algorithms with oracles beyond the linear setting: to submodular [18] and non-convex [2] settings.

In bandit learning, many interesting results and powerful optimization/algorithmic methods have been proved and introduced, including interior point methods [1], random walk [26], continuous multiplicative updates [9], random perturbation [3], iterative methods [13]. In bandit linear optimization, the near-optimal regret bound of $\widetilde{O}(n\sqrt{T})$ has been established due to a long line of works [1, 9, 6]. Beyond the linear functions, several results have been known. Kleinberg [22], Flaxman et al. [13] provided $\widetilde{O}(\text{poly}(n)T^{3/4})$-regret algorithm for general convex functions. Subsequently, Hazan and Li [19] presented an (exponential-time) algorithm which achieves $\widetilde{O}(\exp(n)\sqrt{T})$-regret. Recently, Bubeck et al. [7] gave the first polynomial-time algorithm with regret $\widetilde{O}(n^{9.5}\sqrt{T})$.

**Smooth Auctions and Fictitious Play.**  The smoothness framework was introduced in order to prove approximation guarantees for equilibria in complete-information [27] and incomplete-information [32, 28] games. Smooth auctions (Definition 2) is a large class of auctions where the price of anarchy can be systematically characterized by the smooth arguments. Many interesting auctions have been shown to be smooth; and the smooth argument is a central proof technique to analyze the price of anarchy. We refer the reader to a recent survey [30] for more details. The smoothness framework extends to adaptive dynamics with vanishing regret. However, several important dynamics are not guaranteed to have the vanishing regret property, for example the class of fictitious play [5] and other classes of dynamics in [15]. A research agenda, as raised in [30], is to characterize the performance of such dynamics. Some recent works (e.g., [24]) have been considered in this direction.

**Revenue Maximization.**  Optimal truthful auctions in single-parameter environments are completely characterized by Myerson [25]. Recently, a major line of research in data-driven mechanism design focus on competitive auctions without the full knowledge on the valuation distribution and even in non-stochastic settings. The study of second-price auctions with reserve prices in single-parameter environments and its variants have been considered in [21, 4, 8]. Recently, Roughgarden and Wang [29] gave a polynomial-time online algorithm that achieves $(1/2, O(\sqrt{T}))$-regret. Subsequently, Dudik et al. [11] showed that the same regret bound can be obtained using their framework. Both are in the online full-information setting.

**No-envy Learning in Auctions.**  The notion of *no-envy learning* in auctions has been introduced by Daskalakis and Syrgkanis [10]. They proposed the concept of no-envy learning in order to maintain both the welfare optimality and computational tractability. Among others, Daskalakis and Syrgkanis [10] considered the online item selection problem with coverage valuation and gave an efficient $(1 - 1/e)$-approximate no-envy algorithm with regret bound of $O(\sqrt{T})$.

## 1.4   Organization

Due to the space limit, we present only the revenue maximization (description in Section 1.2.2) as an application. We refer the reader to the supplementary for the full paper with all applications (and complete proofs).

## 2   Framework of Online Learning

We present a general efficient online algorithm and characterize its regret bound based on its concavity parameters. In Section 2.1, we prove the guarantee of the online mirror descent algorithm assuming access to unbiased estimates of the gradients of the functions. In Section 2.2, we derive an algorithm in the bandit setting.

### 2.1   Regret of $(\lambda, \mu)$-Concave Functions

**Mirror descent.**  We are given a convex set $\mathcal{K}$. Let $\Phi$ be a $\alpha_\Phi$-strongly convex function w.r.t a norm $\|\cdot\|$. (A function $\Phi : \mathbb{R}^n \to \mathbb{R}$ is $\alpha_\Phi$-*strongly convex* w.r.t $\|\cdot\|$ if $\Phi(\boldsymbol{x}') \geq \Phi(\boldsymbol{x}) + \langle \nabla\Phi(\boldsymbol{x}), \boldsymbol{x}' - \boldsymbol{x}\rangle + \frac{\alpha_\Phi}{2}\|\boldsymbol{x}' - \boldsymbol{x}\|^2$.) Initially, let $\boldsymbol{x}^1$ be an arbitrary point in $\mathcal{K}$. At time step $t$, play $\boldsymbol{x}^t$ and receive the reward of $F^t(\boldsymbol{x}^t)$. Let $\boldsymbol{g}^t$ be the unbiased estimate of $-\nabla F^t(\boldsymbol{x}^t)$ revealed at time $t$. The algorithm selects the

decision $\boldsymbol{x}^{t+1}$ using the standard mirror descent: $\boldsymbol{x}^{t+1} = \arg\max_{\boldsymbol{x}\in\mathcal{K}}\big\{\langle\eta\boldsymbol{g}^t, \boldsymbol{x}-\boldsymbol{x}^t\rangle - D_\Phi(\boldsymbol{x}\|\boldsymbol{x}^t)\big\}$. where the *Bregman divergence* is defined as $D_\Phi(\boldsymbol{x}\|\boldsymbol{x}') := \Phi(\boldsymbol{x}) - \Phi(\boldsymbol{x}') - \langle\nabla\Phi(\boldsymbol{x}'), \boldsymbol{x}-\boldsymbol{x}'\rangle$.

**Theorem 1** *Assume that $F^t$ is $(\lambda, \mu)$-concave for every $1 \le t \le T$ and $\boldsymbol{x}^*$ is the best solution in hindsight, i.e., $\boldsymbol{x}^* \in \arg\max_{\mathcal{K}}\sum_{t=1}^T F^t(\boldsymbol{x})$. Then the mirror descent algorithm achieves $\big(\frac{\lambda}{\mu}, R(T)\big)$-regret in expectation where $R(T) = \frac{1}{\mu\cdot\eta}D_\Phi(\boldsymbol{x}^*\|\boldsymbol{x}^1) + \frac{\eta}{\mu\cdot 2\alpha_\Phi}\sum_{t=1}^T \|\boldsymbol{g}^t\|_*^2$. If $\|\boldsymbol{g}^t\|_* \le L_g$ for $1 \le t \le T$ (i.e., $F^t$ is $L_g$-Lipschitz w.r.t $\|\cdot\|$) and $D_\Phi(\boldsymbol{x}^*\|\boldsymbol{x}^1)$ is bounded by $G^2$ then by choosing $\eta = \frac{G}{L_g}\sqrt{\frac{2\alpha_\Phi}{T}}$, we have $R(T) \le \frac{GL_g}{\mu}\sqrt{2\alpha_\Phi T}$.*

## 2.2 Bandit Algorithm

In this section, we consider the bandit setting in which at every time $t$ one can observe only the reward $f^t(\boldsymbol{x}^t)$ where $f^t$ is a bounded function defined on the convex set $\mathcal{K} = [0,1]^n$. Without loss of generality, assume that $f^t : [0,1]^n \to [0,1]$. In our algorithm, we will consider a discretization of $[0,1]^n$ and the multilinear relaxations of functions $f^t$ on these discrete points constructed as follows.

**Discretization and Multilinear Extension.** Let $f : [0,1]^n \to [0,1]$ be a function. Consider a lattice $\mathcal{L} = \{0, 2^{-M}, 2\cdot 2^{-M}, \ldots, \ell\cdot 2^{-M}, \ldots, 1\}^n$ where $0 \le \ell \le 2^M$ for some large parameter $M$ as a discretization of $[0,1]^n$. $M$ is a constant parameter to be chosen later. Note that each $x_i \in \{0, 2^{-M}, 2\cdot 2^{-M}, \ldots, \ell\cdot 2^{-M}, \ldots, 1\}$ can be uniquely decomposed as $x_i = \sum_{j=0}^M 2^{-j}y_{ij}$ where $y_{ij} \in \{0,1\}$. By this observation, we lift the set $[0,1]^n \cap \mathcal{L}$ to the $(n\times(M+1))$-dim space. Specifically, define a bijective *lifting* map $m : [0,1]^n \cap \mathcal{L} \to \{0,1\}^{n\times(M+1)}$ such that each point $(x_1, \ldots, x_n) \in \mathcal{K} \cap \mathcal{L}$ is mapped to the unique point $(y_{10}, \ldots, y_{1M}, \ldots, y_{n0}, \ldots, y_{nM}) \in \{0,1\}^{n\times(M+1)}$ where $x_i = \sum_{j=0}^M 2^{-j}y_{ij}$. Define function $\tilde{f} : \{0,1\}^{n\times(M+1)} \to [0,1]$ such that $\tilde{f}(\mathbf{1}_S) := f(m^{-1}(\mathbf{1}_S))$; in other words, $\tilde{f}(\mathbf{1}_S) = f(\boldsymbol{x})$ where $\boldsymbol{x} \in [0,1]^n \cap \mathcal{L}$ and $\mathbf{1}_S = m(\boldsymbol{x})$. Note that $\mathbf{1}_S$ with $S \subset [n\times(M+1)]$ is a $(n\times(M+1))$-dim vector with $(ij)^{th}$-coordinate equal to 1 if $(i,j) \in S$ and equal to 0 otherwise. Consider a multilinear extension $F : [0,1]^{n\times(M+1)} \to [0,1]$ of $\tilde{f}$ defined as follows.

$$F(\boldsymbol{z}) := \sum_{S \subset [n\times(M+1)]} \tilde{f}(\mathbf{1}_S) \prod_{(i,j)\in S} z_{ij} \prod_{(i,j)\notin S} (1 - z_{ij}).$$

By the definition, $F(\boldsymbol{z})$ can be seen as $\mathbb{E}[\tilde{f}(\mathbf{1}_S)]$ where the $(ij)^{th}$-coordinate of $\mathbf{1}_S$ equals 1 (i.e., $(\mathbf{1}_S)_{ij} = 1$) with probability $z_{ij}$.

**Algorithm description.** Our algorithm, formally given in Algorithm 1, is inspired by algorithm SCRIBLE [1] which has been derived in the context of bandit linear optimization. It has been observed that the gradient estimates of the functions in SCRIBLE are unbiased only if those functions are linear; and that represents a main obstacle in order to derive an algorithm with optimal regret guarantee $R(T) = \widetilde{O}(\sqrt{T})$. While aiming for the regret of $\widetilde{O}(\sqrt{T})$, in our algorithm, we overcome this obstacle by considering at every step the gradient estimate of the multilinear extension of the reward function (construction above). That gradient estimate will be indeed proved to be unbiased. Incorporating that gradient estimator to the scheme in [1] and following our approach, we prove the regret guarantee of the algorithm. Note that in our algorithm, we do not need the information about the concavity parameters of the functions.

**Theorem 2** *Let $f^t : [0,1]^n \to [0,1]$ be the reward function at time $1 \le t \le T$ and let $F^t$ be the multilinear extension of the discretization of $f^t$ based on a lattice $\mathcal{L}$ (defined earlier). Assume that $F^t$'s are $(\lambda, \mu)$-concave and for every $\boldsymbol{x} \in [0,1]^n$, there exists $\overline{\boldsymbol{x}} \in \mathcal{L}$ such that $|f^t(\boldsymbol{x}) - f^t(\overline{\boldsymbol{x}})| \le L \cdot 2^{-M}$ for every $1 \le t \le T$ (for example, $f^t$'s are $L$-Lipschitz). Then, by choosing $M = \log T$ and $\eta = O\big(\frac{1}{(nM)^{3/2}\cdot\sqrt{T}}\big)$ and $\Phi$ as a $O(nM)$-self-concordant function, Algorithm 1 (mirroir descent algorithm) achieves:*

$$\sum_{t=1}^T \mathbb{E}\big[f^t(\boldsymbol{x}^t)\big] \ge \frac{\lambda}{\mu} \cdot \max_{\boldsymbol{x}\in[0,1]^n}\sum_{t=1}^T f^t(\boldsymbol{x}) - O\big(\max\{\lambda/\mu, 1\}Ln^{3/2}(\log T)^{3/2}(\log\log T)\sqrt{T}\big).$$

**Algorithm 1** Algorithm in the bandit setting.

1: Let $\Phi$ be a $\nu$-self-concordant function over $[0,1]^{n\times(M+1)}$.
2: Let $\boldsymbol{z}^1 \in \text{int}([0,1]^{n\times(M+1)})$ such that $\nabla\Phi(\boldsymbol{z}^1) = 0$.
3: **for** $t = 1$ to $T$ **do**
4:     Let $\boldsymbol{A}^t = \left[\nabla^2\Phi(\boldsymbol{z}^t)\right]^{-1/2}$.
5:     Pick $\boldsymbol{u}^t \in \mathbb{S}_n$ uniformly random and set $\boldsymbol{y}^t = \boldsymbol{z}^t + \boldsymbol{A}^t\boldsymbol{u}^t$.
6:     Round $\boldsymbol{y}^t$ to a random point $\boldsymbol{1}_{S^t} \in \{0,1\}^{n\times(M+1)}$ such that element $(i,j)$ appears in $S^t$ with probability $y_{ij}^t$.
7:     Play $\boldsymbol{x}^t = m^{-1}(\boldsymbol{1}_{S^t})$ and receive the reward of $f^t(\boldsymbol{x}^t)$.
8:     Let $\boldsymbol{g}^t = -n(M+1)f^t(\boldsymbol{x}^t)(\boldsymbol{A}^t)^{-1}\boldsymbol{u}^t$ and compute the solution $\boldsymbol{z}^{t+1} \in [0,1]^{n\times(M+1)}$ by applying the mirror descent framework on $F^t$ (Section 2.1). Specifically,

$$\boldsymbol{z}^{t+1} = \arg\max_{\boldsymbol{z}\in[0,1]^{n\times(M+1)}} \left\{\langle \eta\boldsymbol{g}^t, \boldsymbol{z} - \boldsymbol{z}^t\rangle - D_\Phi(\boldsymbol{z}\|\boldsymbol{z}^t)\right\}.$$

## 3 Online Simultaneous Second-Price Auctions with Reserve Prices

In this section, we consider the online simultaneous second-price auctions with reserve prices (definition in Section 1.2.2). We denote the revenue of selling item $j$ as $\text{REV}_j(\boldsymbol{r}^t, \boldsymbol{b}^t)$ where $\boldsymbol{b}^t = (b_1^t, \ldots, b_n^t)$ and $\boldsymbol{r}^t = (r_1^t, \ldots, r_n^t)$. The revenue of the auctioneer at time step $t$ is $\text{REV}(\boldsymbol{r}^t, \boldsymbol{b}^t) = \sum_{j=1}^m \text{REV}_j(\boldsymbol{r}^t, \boldsymbol{b}^t)$. The goal of the auctioneer is to achieve the total revenue $\sum_{t=1}^T \text{REV}(\boldsymbol{r}^t, \boldsymbol{b}^t)$ approximately close to that achieved by the best fixed reserve-price $\max_{\boldsymbol{r}^*} \sum_{t=1}^T \text{REV}(\boldsymbol{r}^*, \boldsymbol{b}^t)$.

In the setting, by scaling, assume that all bids and reserve prices are in $\mathcal{K} = [0,1]^{n\times m}$. Consider the lattice $\mathcal{L} = \{\ell\cdot 2^{-M} : 0 \leq \ell \leq 2^M\}^{n\times m} \subset [0,1]^{n\times m}$ for some large parameter $M$ as a discretization of $[0,1]^{n\times m}$. Observe that for any reserve price vector $\boldsymbol{r}$, $|\text{REV}(\boldsymbol{r},\boldsymbol{b}) - \text{REV}(\overline{\boldsymbol{r}},\boldsymbol{b})| \leq m\cdot 2^{-M}$ where $\overline{\boldsymbol{r}}$ is a reserve price vector such that $\overline{r}_{ij}$ is the largest multiple of $2^{-M}$ smaller than $r_{ij}$ for every $i,j$ (for some large enough $M$). Therefore, one can approximate the revenue up to any arbitrary precision by restricting the reserve price on $\mathcal{L}$. We slightly abuse notation by denoting $\text{REV}_j(\boldsymbol{1}_S, \boldsymbol{b})$ as $\text{REV}_j(\boldsymbol{r}, \boldsymbol{b})$ where $\boldsymbol{1}_S = m(\boldsymbol{r})$ (recall that $m$ is the map defined in Section 2.2). Following Section 2.2, given a bid vector $\boldsymbol{b}$, the multilinear extension $\overline{\text{REV}}$ of the revenue $\text{REV}$ is defined as $\overline{\text{REV}}(\cdot, \boldsymbol{b}) : [0,1]^{n\times m\times(M+1)} \to \mathbb{R}$ such that:

$$\overline{\text{REV}}(\boldsymbol{z}, \boldsymbol{b}) = \sum_{S\subset[n\times m\times(M+1)]} \left(\sum_{j=1}^m \text{REV}_j(\boldsymbol{1}_S, \boldsymbol{b})\right) \prod_{(i,j,k)\in S} z_{ijk} \prod_{(i,j,k)\notin S} (1 - z_{ijk}).$$

**Online bandit Reserve-Price Algorithm.** Initially, let $\boldsymbol{r}^1$ be an arbitrary feasible reserve-price. At each time step $t \geq 1$,

(i) select $\boldsymbol{r}^t$ or $\boldsymbol{0}$ each with probability $1/2$ as the reserve-price;

(ii) receive the revenue corresponding to the selected reserve-price;

(iii) compute $\boldsymbol{r}^{t+1}$ using Algorithm 1 with the following specification: in line 8 of Algorithm 1, replace $f^t(\boldsymbol{x}^t)$ by $2\text{REV}(\boldsymbol{r}^t, \boldsymbol{b}^t)$ if the selected reserve-price is $\boldsymbol{r}^t$, or replace $f^t(\boldsymbol{x}^t)$ by $0$ if the selected reserve-price is $\boldsymbol{0}$. (By doing that, the expected value of $\boldsymbol{g}^t$ in Algorithm 1 is $-\nabla\overline{\text{REV}}(\boldsymbol{r}^t, \boldsymbol{b}^t)$.)

**Analysis.** In order to analyze the performance of this algorithm, we study the properties of some related functions and then derive the regret bound for the algorithm. Fix a bid vector $\boldsymbol{b}$. Let $\boldsymbol{r}_j$ be a vector consisting of reserve prices on item $j$, i.e., $\boldsymbol{r}_j = (r_{1j}, \ldots, r_{nj})$. (Recall that $r_{ij}$ is the reserve price for bidder $i$ on item $j$.) As $\boldsymbol{b}$ is fixed and the selling procedure of each item depends only on the reserve prices to the item, so for simplicity denote $\text{REV}_j(\boldsymbol{r}, \boldsymbol{b})$ as $\text{REV}_j(\boldsymbol{r}_j)$ and $\text{REV}(\boldsymbol{r}, \boldsymbol{b})$ as $\text{REV}(\boldsymbol{r})$. Define a function $h_j : \{0,1\}^{n\times(M+1)} \to \mathbb{R}$ such that $h_j(\boldsymbol{1}_T) = \max\{\text{REV}_j(\boldsymbol{1}_T), \text{REV}_j(\boldsymbol{1}_\emptyset)\} = \max\{\text{REV}_j(\boldsymbol{r}), \text{REV}_j(\boldsymbol{0})\}$ where $\boldsymbol{r}_j$ is the reserve price corresponding to $\boldsymbol{1}_T$ for $T \subset [n \times (M+1)]$. Let $H_j : [0,1]^{n\times(M+1)} \to \mathbb{R}$ be the multilinear extension of $h_j$. Moreover, define $H :$

$[0,1]^{n \times m \times (M+1)} \to \mathbb{R}$ as the multilinear extension of $\max\{\text{REV}(\boldsymbol{r}), \text{REV}(\boldsymbol{0})\}$ defined as $H(\boldsymbol{z}) = \sum_{S \subset [n \times m \times (M+1)]} \max\{\text{REV}(\boldsymbol{1}_S), \text{REV}(\boldsymbol{1}_\emptyset)\} \prod_{(i,j,k) \in S} z_{ijk} \prod_{(i,j,k) \notin S}(1 - z_{ijk})$.

**Lemma 1** *It holds that $H(\boldsymbol{z}) = \sum_{j=1}^{m} H_j(\boldsymbol{z}_j)$ where $\boldsymbol{z}_j$ is the restriction of $\boldsymbol{z}$ to the coordinate related to item $j$.*

**Lemma 2** *Function $H_j$ is $(1,1)$-concave.*

*Proof* We prove that the condition $(1)$ of the $(1,1)$-concavity holds for all points in the lattice. As the multilinear extension can be seen as the expectation over these points, the lemma will follow. Fix a bid profile $\boldsymbol{b}_j = (b_{1j}, \ldots, b_{nj})$. Without loss of generality, assume that $b_{1j} \geq b_{2j} \geq \ldots \geq b_{nj}$. Let $\boldsymbol{r}_j$ and $\boldsymbol{r}_j^*$ be two arbitrary reserve price vectors. We will show that

$$\sum_{i=1}^{n} \left[ \max\{\text{REV}_j(\boldsymbol{r}_{-i,j}, r_{ij}^*), \text{REV}_j(\boldsymbol{0})\} - \max\{\text{REV}_j(\boldsymbol{r}_j), \text{REV}_j(\boldsymbol{0})\} \right]$$
$$\geq \max\{\text{REV}_j(\boldsymbol{r}_j^*), \text{REV}_j(\boldsymbol{0})\} - \max\{\text{REV}_j(\boldsymbol{r}_j), \text{REV}_j(\boldsymbol{0})\} \quad (2)$$

where $\boldsymbol{r}_{-ij}$ stands for the reserve price vectors on item $j$ without the reserve price of bidder $i$.

Observe that the revenue $\max\{\text{REV}_j(\boldsymbol{r}_j'), \text{REV}_j(\boldsymbol{0})\}$ for every reserve price $\boldsymbol{r}_j'$ is at least the second highest bid $b_{2j}$ (that is obtained in $\text{REV}_j(\boldsymbol{0})$). Moreover, for any reserve price $\boldsymbol{r}_j'$ such that the auctioneer either $(1)$ removes the first bidder (with highest bid) or $(2)$ removes the second bidder and $r_{1j}' \leq b_{2j}$, the revenue $\max\{\text{REV}_j(\boldsymbol{r}_j'), \text{REV}_j(\boldsymbol{0})\} = \text{REV}_j(\boldsymbol{0})$. Hence, $\max\{\text{REV}_j(\boldsymbol{r}_j'), \text{REV}_j(\boldsymbol{0})\} \neq \text{REV}_j(\boldsymbol{0})$ if and only if $b_{2j} < r_{1j}' \leq b_{1j}$.

By these observations, we deduce that $\max\{\text{REV}_j(\boldsymbol{r}_{-ij}, r_{ij}^*), \text{REV}_j(\boldsymbol{0})\} \neq \max\{\text{REV}_j(\boldsymbol{r}_j), \text{REV}_j(\boldsymbol{0})\}$ if and only if $i = 1$ and either $\{b_{2j} \leq r_{1j} \neq r_{1j}^* \leq b_{1j}\}$; or $\{r_{1j}^* \in (b_{2j}, b_{1j}] \text{ and } r_{1j} \notin (b_{2j}, b_{1j}]\}$; or inversely $\{r_{1j} \in (b_{2j}, b_{1j}] \text{ and } r_{1j}^* \notin (b_{2j}, b_{1j}]\}$.

Thus, proving Inequality $(2)$ is equivalent to showing that

$$\max\{\text{REV}_j(\boldsymbol{r}_{-1j}, r_{1j}^*), \text{REV}_j(\boldsymbol{0})\} - \max\{\text{REV}_j(\boldsymbol{r}_j), \text{REV}_j(\boldsymbol{0})\}$$
$$\geq \max\{\text{REV}_j(\boldsymbol{r}_j^*), \text{REV}_j(\boldsymbol{0})\} - \max\{\text{REV}_j(\boldsymbol{r}_j), \text{REV}_j(\boldsymbol{0})\}.$$

**Case** 1: $b_{2j} \leq r_{1j} \neq r_{1j}^* \leq b_{1j}$. In this case, both sides are equal to $r_{1j}^* - r_{1j}$.

**Case** 2: $r_{1j}^* \in (b_{2j}, b_{1j}]$ and $r_{1j} \notin (b_{2j}, b_{1j}]$. In this case, both sides are equal to $r_{1j}^* - b_{2j}$.

**Case** 3: $r_{1j} \in (b_{2j}, b_{1j}]$ and $r_{1j}^* \notin (b_{2j}, b_{1j}]$. In this case, both sides are equal to $b_{2j} - r_{1j}$.

**Case** 4: the complementary of all previous cases. In this case, both sides are equal to $0$.

Therefore, Inequality $(2)$ holds and so the lemma follows. $\qquad\square$

Consider an imaginary algorithm which is similar to our online reserve price algorithm but at every step $t$, its gain on item $j$ is $\max\{\text{REV}_j(\boldsymbol{r}^t), \text{REV}_j(\boldsymbol{0})\}$. Observe that the online reserve price algorithm selects at every step $t$ either $\boldsymbol{r}^t$ or $\boldsymbol{0}$ with probability $1/2$, the revenue of the algorithm is at least half that of the imaginary algorithm. Hence, by Theorem 2 and the $(1,1)$-concavity of $H$ (by Lemmas 1 and 2), we deduce the following theorem.

**Theorem 3** *The online bandit reserve price algorithm achieves $\left(1/2, O(m\sqrt{nm} \log T \sqrt{T})\right)$-regret.*

## 4 Conclusion

In this paper, we have introduced a framework to design efficient online learning algorithms. Apart of standard regularity requirements (such as compact convex domain, Lipschitz, etc), a new crucial property is the $(\lambda, \mu)$-concavity. Designing efficient online learning algorithms is now reduced to determining the concave parameters of reward functions. We show the applicability of the framework through applications in auction design. Due to the simplicity of the new notion of concavity, we hope that our approach would be useful in designing efficient online algorithms with approximate regret bounds for different problems.

## Broader Impact

As for the ethical and future societal direct consequences, this is not relevant in the context of this paper.

## Acknowledgments and Disclosure of Funding

This work is supported by the ANR project OATA n° ANR-15-CE40-0015-01.

## Footnotes

[1]A coverage function $v : 2^{[m]} \to \mathbb{R}^+$ has the form $v(S) = |\cup_{j \in S} A_j|$ where $A_1, \ldots, A_m$ are subsets of $[m]$.

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
