[Supplementary Material]

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

(a), u_i(a; v_i), p_i(a), \text{Sw}(a; v), \text{Rev}(a, v)$ instead of $v_i(o(a)), u_i(o(a); v_i), p_i(o(a)), \text{Sw}(o(a); v), \text{Rev}(o(a), v)$, respectively. In the paper, we consider two standard objectives: welfare maximization and revenue maximization. Note that in revenue maximization, we call players as bidders.

### 1.2.1 Fictitious Play in Smooth Auctions

We consider adaptive dynamics in auctions. In the setting, there is an underlying auction $o$ and there are $n$ players, each player $i$ has a set of actions $\mathcal{A}_i$ and a valuation function $v_i$ taking values in $[0, 1]$ (by normalization). In every time step $1 \le t \le T$, each player $i$ selects a strategy which is a distribution in $\Delta(\mathcal{A}_i)$ according to some given adaptive dynamic. After all players have committed their strategies, which result in a strategy profile $\sigma^t \in \Delta(\mathcal{A})$, the auction induces a social welfare $\text{Sw}(o, \sigma^t) := \mathbb{E}_{a \sim \sigma^t}[\text{Sw}(o(a); v)]$. In this setting, we study the total welfare achieved by the given adaptive dynamic comparing to the optimal welfare. This problem can be cast in the online optimization framework in which at time step $t$, the player strategy profile corresponds to the decision of the algorithm and subsequently, the gain of the algorithm is the social welfare induced by the auction w.r.t the strategy profile.

Smooth auctions is an important class of auctions in welfare maximization. The smoothness notion has been introduced [40, 36] in order to characterize the efficiency of (Bayes-Nash) equilibria of auctions. It has been shown that several auctions in widely studied settings are smooth; and many proof techniques analyzing equilibrium efficiency can be reduced to the smooth argument.

**Definition 3 ([40, 36])** *For parameters $\lambda, \mu \ge 0$, an auction is $(\lambda, \mu)$-smooth if for every valuation profile $v = (v_1, \ldots, v_n)$, there exist action distributions $\overline{D}_1(v), \ldots, \overline{D}_n(v)$ over $\mathcal{A}_1, \ldots, \mathcal{A}_n$ such that, for every action profile $a$,*

$$\sum_{i=1}^n \mathbb{E}_{\overline{a}_i \sim \overline{D}_i(v)}[u_i(\overline{a}_i, a_{-i}; v_i)] \ge \lambda \cdot \text{Sw}(\overline{a}; v) - \mu \cdot \text{Sw}(a; v)$$

*where $a_{-i}$ is the action profile similar to $a$ without player $i$.*

It has been proved that if an auction is $(\lambda, \mu)$-smooth then every Bayes-Nash equilibrium of the auction has expected welfare at least $\lambda/\mu$ fraction of the optimal auction [36, 40]. The performance guarantee holds even for vanishing regret sequences. A sequence of actions profiles $a^1, a^2, \ldots,$ is an *individually-vanishing-regret sequence* if for every player $i$ and action $a_i'$,

$$\lim_T \frac{1}{T} \sum_{t=1}^T [u_i(a_i', a_{-i}^t; v_i) - u_i(a^t; v_i)] \le 0. \tag{2}$$

However, several interesting dynamics are not guaranteed to have the individually-vanishing-regret property. In a recent survey, Roughgarden et al. [38] have raised a question whether adaptive dynamics without the individually-vanishing-regret condition can achieve approximate optimal welfare. Among others, *fictitious play* [8] is an interesting, widely-studied dynamic which attracts a significant attention in the community.

In the paper, we consider a version of fictitious play in smooth auctions, namely Perturbed Discrete Time Fictitious Play (PDTFP). (The formal definition is given in Section 4.) In general, it is not known whether this dynamic has individually-vanishing-regret. (In particular, in the PDTFP dynamic players are valuation-oriented whereas the condition (2) concerns player utilities.) Despite that fact, using our framework, we prove that given an offline $(\lambda, \mu)$-smooth auction, PDTFP dynamic achieves a $\lambda/(1 + \mu)$ fraction of the optimal welfare. The corresponding formal statement is Theorem 3.

**Informal Theorem 2** *If the underlying auction $o$ is a $(\lambda, \mu)$-smooth then the PDTFP dynamic achieves $\left(\frac{\lambda}{1+\mu}, R(T)\right)$-regret where $R(T) = O\left(\frac{\sqrt{T}}{1+\mu}\right)$.*

### 1.2.2 Revenue maximization in Multi-Dimensional Environments

We consider online simultaneous second-price auctions with reserve prices in *multi-dimensional* environments. In the setting, there are $n$ bidders and $m$ items to be sold to these bidders. At every time step $t = 1, 2, \ldots, T$, the auctioneer selects reserve prices $r_i^t = (r_{i1}^t, \ldots, r_{im}^t)$ for each bidder $i$ where $r_{ij}^t$ is the reserve price of item $j$ for bidder $i$. Each bidder $i$ for $1 \le i \le n$ has a (private) valuation $v_i^t : 2^{[m]} \to \mathbb{R}^+$ over subsets of items. After the reserve prices have been chosen, every bidder $i$ picks a bid vector $b_i^t$ where $b_{ij}^t$ is the bid of bidder $i$ on item $j$ for $1 \le j \le m$. Then the auction for each item $1 \le j \le m$ works as follows: (1) remove all bidders $i$ with $b_{ij}^t < r_{ij}^t$; (2) run the second price auction on the remaining bidders to determine the winner of item $j$ — the bidder with highest non-removed bid on item $j$; and (3) charge the winner of item $j$ the price which is the maximum of $r_{ij}^t$ and the second highest bid among non-removed bids $b_{ij}^t$. The objective of the auctioneer is to achieve the total revenue approximately close to that achieved by the best fixed reserve-price auction. Note that in the bandit setting, the auction is given as a blackbox and at every time step, the auctioneer observes only the total revenue (total price) without knowing neither the bids of bidders nor the winner/price of each item. The setting enhances, among others, the privacy of bidders.

The second-price auctions with reserve prices in *single-parameter* environments have been considered by Roughgarden and Wang [37] in full-information online learning. Using the Follow-the-Perturbed-Leader strategy, they gave a polynomial-time online algorithm that achieves half the revenue of the best fixed reserve-price auction minus a term $O(\sqrt{T} \log T)$ (so their algorithm is $(1/2, O(\sqrt{T} \log T))$-regret in our terminology). The problem we consider cannot be reduced to applying their algorithm on $m$ separate items since (1) bids on different items might be highly correlated (due to bidders' valuations); and (2) in the bandit setting for multiple items, the auctioneer know only the total revenue (not the revenue from each item). Using our framework, we prove the following result.

**Informal Theorem 3** *There exist a bandit algorithm that achieves the regret bound of* $\left(1/2, O\left(mn^{3/2}(\log T)^{3/2}(\log \log T)\sqrt{T}\right)\right)$ *for

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

We begin by giving some preliminary definitions in Section 2. In Section 3, we present our framework and the main algorithm. Subsequently, we show the applications about: (1) Perturbed Fictitious Play in Smooth Auctions in Section 4; (2) Online Simultaneous Second-Price Auctions with Reserve Prices in Section 5; and (3) Bandit No-Envy Learning in Auctions in Section 6.

## 2 Preliminaries

Given a norm $\|\cdot\|$, the dual norm is defined as $\|\boldsymbol{y}\|_* := \max_{\boldsymbol{x}:\|\boldsymbol{x}\|=1} \langle \boldsymbol{x}, \boldsymbol{y} \rangle$. A function $\Phi : \mathbb{R}^n \to \mathbb{R}$ is $\alpha_\Phi$-*strongly convex* w.r.t $\|\cdot\|$ if

$$\Phi(\boldsymbol{x}') \geq \Phi(\boldsymbol{x}) + \langle \nabla\Phi(\boldsymbol{x}), \boldsymbol{x}' - \boldsymbol{x} \rangle + \frac{\alpha_\Phi}{2} \|\boldsymbol{x}' - \boldsymbol{x}\|^2$$

Given a strongly convex function $\Phi$, a map defined as $\boldsymbol{x} \mapsto \nabla\Phi(\boldsymbol{x})$ is bijective. Denote $\nabla\Phi^*$ the inverse map of $\nabla\Phi$. In fact, this inverse map is given by the gradient of the Fenchel dual for $\Phi$. We refer reader to [5] and [6, Chapter 7] for more details. Given a strictly convex function $\Phi : \mathbb{R}^n \to \mathbb{R}$, the *Bregman divergence* is defined as

$$D_\Phi(\boldsymbol{x}\|\boldsymbol{x}') := \Phi(\boldsymbol{x}) - \Phi(\boldsymbol{x}') - \langle \nabla\Phi(\boldsymbol{x}'), \boldsymbol{x} - \boldsymbol{x}' \rangle$$

The following lemma generalizes the Pythagoras theorem (proof can be found in [4] for example).

**Lemma 1 (Generalized Pythagorean Property)** *Given a convex body $\mathcal{K} \subset \mathbb{R}^n$. Let $\boldsymbol{x} \in \mathcal{K}$ and $\boldsymbol{y}' \in \mathbb{R}^n$. Let $\boldsymbol{y}$ be the projection of $\boldsymbol{y}'$ on $\mathcal{K}$, defined as $\boldsymbol{y} = \arg\min_{\overline{\boldsymbol{y}} \in \mathcal{K}} D_\Phi(\overline{\boldsymbol{y}}\|\boldsymbol{y}')$. Then $D_\Phi(\boldsymbol{x}\|\boldsymbol{y}) \leq D_\Phi(\boldsymbol{x}\|\boldsymbol{y}')$.*

Let $\mathcal{K} \subset \mathbb{R}^n$ be a convex set with non-empty interior $\text{int}(\mathcal{K})$. A function $\Phi : \mathcal{K} \to \mathbb{R}$ is $\nu$-*self-concordant* if

1. $\Phi$ is three times continuously differentiable, convex and $\Phi$ approaches infinity along any sequence approaching the boundary of $\mathcal{K}$

2. For every $\boldsymbol{a} \in \mathcal{R}^n$ and $\boldsymbol{x} \in \text{int}(K)$, it holds that

$$\left| \nabla^3\Phi(\boldsymbol{x})[\boldsymbol{a}, \boldsymbol{a}, \boldsymbol{a}] \right| \leq 2 \big( |\nabla^2\Phi(\boldsymbol{x})[\boldsymbol{a}, \boldsymbol{a}]| \big)^{3/2}$$

$$\left| \nabla\Phi(\boldsymbol{x})[\boldsymbol{a}] \right| \leq \nu^{1/2} \big( |\nabla^2\Phi(\boldsymbol{x})[\boldsymbol{a}, \boldsymbol{a}]| \big)^{1/2}$$

where $\nabla^3\Phi(\boldsymbol{x})[\boldsymbol{a}, \boldsymbol{a}, \boldsymbol{a}] := \frac{\partial^3}{\partial t_1 \partial t_2 \partial t_3} \Phi(\boldsymbol{x} + t_1\boldsymbol{a} + t_2\boldsymbol{a} + t_3\boldsymbol{a}) \Big|_{t_1=t_2=t_3=0}$

One can define a local norm based on the Hessian of a self-concordant function. Formally, given a $\nu$-self-concordant function $\Phi$ and a point $\boldsymbol{x} \in \text{int}(\mathcal{K})$, the local norm $\|\cdot\|_{\boldsymbol{x}}$ and its dual norm $\|\cdot\|_{*,\boldsymbol{x}}$ are defined as

$$\|\boldsymbol{a}\|_{\boldsymbol{x}} = \big( \boldsymbol{a}^\top \nabla^2\Phi(\boldsymbol{x})\boldsymbol{a} \big)^{1/2}, \qquad \|\boldsymbol{a}\|_{*,\boldsymbol{x}} = \big( \boldsymbol{a}^\top (\nabla^2\Phi(\boldsymbol{x}))^{-1}\boldsymbol{a} \big)^{1/2}$$

The following lemma states a useful property of self-concordant functions.

**Lemma 2 ([23], Lemma 6.10)** *Let $\Phi$ be a $\nu$-self-concordant function over a convex set $\mathcal{K}$. Then, for all $\boldsymbol{x}, \boldsymbol{y} \in int(\mathcal{K})$,*

$$\Phi(\boldsymbol{x}) - \Phi(\boldsymbol{y}) \leq \nu \log \frac{1}{1 - \pi_{\boldsymbol{x}}(\boldsymbol{y})}$$

*where $\pi$ is the Minkowski function over $\mathcal{K}$ defined as $\pi_{\boldsymbol{x}}(\boldsymbol{y}) := \inf\{t \geq 0 : \boldsymbol{x} + t^{-1}(\boldsymbol{y} - \boldsymbol{x}) \in \mathcal{K}\}$.*

## 3 Framework of Online Learning

We present a general efficient online algorithm and characterize its regret bound based on its concavity parameters. In Section 3.1, we prove the guarantee of the online mirror descent algorithm assuming access to unbiased estimates of the gradients of the functions. In Section 3.2, we derive an algorithm in the bandit setting.

### 3.1 Regret of $(\lambda, \mu)$-Concave Functions

**Mirror descent.** Given a convex set $\mathcal{K}$. Let $\Phi$ be a $\alpha_\Phi$-strongly convex function w.r.t $\|\cdot\|$. Initially, let $\boldsymbol{x}^1$ is an arbitrary point in $\mathcal{K}$. At time step $t$, play $\boldsymbol{x}^t$ and receive the reward of $F^t(\boldsymbol{x}^t)$. Let $\boldsymbol{g}^t$ be an unbiased estimate of $-\nabla F^t(\boldsymbol{x}^t)$ and denote $\boldsymbol{\theta}^t = \nabla \Phi(\boldsymbol{x}^t)$. The algorithm selects the decision $\boldsymbol{x}^{t+1}$ as follows.

$$\boldsymbol{\zeta}^{t+1} = \boldsymbol{\theta}^t - \eta \cdot \boldsymbol{g}^t$$
$$\boldsymbol{y}^{t+1} = \nabla \Phi^*(\boldsymbol{\zeta}^{t+1})$$
$$\boldsymbol{x}^{t+1} = \arg\min_{\boldsymbol{x} \in \mathcal{K}} D_\Phi(\boldsymbol{x} \| \boldsymbol{y}^{t+1})$$

where $\eta$ is a step size. An equivalent description is

$$\boldsymbol{x}^{t+1} = \arg\max_{\boldsymbol{x} \in \mathcal{K}} \{ \langle \eta \boldsymbol{g}^t, \boldsymbol{x} - \boldsymbol{x}^t \rangle - D_\Phi(\boldsymbol{x} \| \boldsymbol{x}^t) \}. \tag{3}$$

**Theorem 1** *Assume that $F^t$ is $(\lambda, \mu)$-concave for every $1 \le t \le T$ and $\boldsymbol{x}^*$ is the best solution in hindsight, i.e., $\boldsymbol{x}^* \in \arg\max \sum_{t=1}^T F^t(\boldsymbol{x})$. Then the mirror descent algorithm achieves $\left(\frac{\lambda}{\mu}, R(T)\right)$-regret in expectation where*

$$R(T) = \frac{1}{\mu \cdot \eta} D_\Phi(\boldsymbol{x}^* \| \boldsymbol{x}^1) + \frac{\eta}{\mu \cdot 2\alpha_\Phi} \sum_{t=1}^T \|\boldsymbol{g}^t\|_*^2$$

*If $\|\boldsymbol{g}^t\|_* \le L_g$ for $1 \le t \le T$ (i.e., $F^t$ is $L_g$-Lipschitz w.r.t $\|\cdot\|$) and $D_\Phi(\boldsymbol{x}^* \| \boldsymbol{x}^1)$ is bounded by $G^2$ then by choosing $\eta = \frac{G}{L_g}\sqrt{\frac{2\alpha_\Phi}{T}}$, we have $R(T) \le \frac{GL_g}{\mu}\sqrt{2\alpha_\Phi T}$.*

*Proof* In the analysis, we follow the potential argument of Bansal and Gupta [4] and derive a bound based on the concavity parameters. Define the potential as $\Psi^t = \frac{1}{\eta} D_\Phi(\boldsymbol{x}^* \| \boldsymbol{x}^t)$. First, we observe that

$$\eta \left( \Psi^{t+1} - \Psi^t \right) = D_\Phi(\boldsymbol{x}^* \| \boldsymbol{x}^{t+1}) - D_\Phi(\boldsymbol{x}^* \| \boldsymbol{x}^t)$$
$$\le D_\Phi(\boldsymbol{x}^* \| \boldsymbol{y}^{t+1}) - D_\Phi(\boldsymbol{x}^* \| \boldsymbol{x}^t)$$
$$= \Phi(\boldsymbol{x}^*) - \Phi(\boldsymbol{y}^{t+1}) - \langle \underbrace{\nabla \Phi(\boldsymbol{y}^{t+1})}_{\boldsymbol{\zeta}^{t+1}}, \boldsymbol{x}^* - \boldsymbol{y}^{t+1} \rangle - \Phi(\boldsymbol{x}^*) + \Phi(\boldsymbol{x}^t) + \langle \underbrace{\nabla \Phi(\boldsymbol{x}^t)}_{\boldsymbol{\theta}^t}, \boldsymbol{x}^* - \boldsymbol{x}^t \rangle$$
$$= \Phi(\boldsymbol{x}^t) - \Phi(\boldsymbol{y}^{t+1}) - \langle \boldsymbol{\zeta}^{t+1}, \boldsymbol{x}^t - \boldsymbol{y}^{t+1} \rangle - \langle \boldsymbol{\zeta}^{t+1} - \boldsymbol{\theta}^t, \boldsymbol{x}^* - \boldsymbol{x}^t \rangle$$
$$= \Phi(\boldsymbol{x}^t) - \Phi(\boldsymbol{y}^{t+1}) - \langle \boldsymbol{\theta}^t, \boldsymbol{x}^t - \boldsymbol{y}^{t+1} \rangle + \langle \eta \boldsymbol{g}^t, \boldsymbol{x}^t - \boldsymbol{y}^{t+1} \rangle + \langle \eta \boldsymbol{g}^t, \boldsymbol{x}^* - \boldsymbol{x}^t \rangle$$
$$\le -\frac{\alpha_\Phi}{2} \|\boldsymbol{y}^{t+1} - \boldsymbol{x}^t\|^2 + \eta \langle \boldsymbol{g}^t, \boldsymbol{x}^t - \boldsymbol{y}^{t+1} \rangle + \eta \langle \boldsymbol{g}^t, \boldsymbol{x}^* - \boldsymbol{x}^t \rangle$$
$$= -\frac{\alpha_\Phi}{2} \|\boldsymbol{y}^{t+1} - \boldsymbol{x}^t\|^2 + \frac{1}{\alpha_\Phi} \langle \eta \boldsymbol{g}^t, \alpha_\Phi \left( \boldsymbol{x}^t - \boldsymbol{y}^{t+1} \right) \rangle + \eta \langle \boldsymbol{g}^t, \boldsymbol{x}^* - \boldsymbol{x}^t \rangle$$
$$\le \frac{\eta^2}{2\alpha_\Phi} \|\boldsymbol{g}^t\|_*^2 + \eta \langle \boldsymbol{g}^t, \boldsymbol{x}^* - \boldsymbol{x}^t \rangle$$

where the first inequality is due to the generalized Pythagorean property (Lemma 1); the fourth equality follows the update rule $\boldsymbol{\zeta}^{t+1} = \boldsymbol{\theta}^t - \eta \cdot \boldsymbol{g}^t$; the second inequality holds since $\Phi$ is $\alpha_\Phi$-strongly convex; and in the last inequality, we use Cauchy-Schwarz inequality $\langle \boldsymbol{a}, \boldsymbol{b} \rangle \le \|\boldsymbol{b}\| \|\boldsymbol{a}\|_* \le \|\boldsymbol{b}\|^2/2 + \|\boldsymbol{a}\|_*^2/2$.

By the observation and the fact that $\boldsymbol{g}^t$ is an unbiased estimate of $-\nabla F^t(\boldsymbol{x}^t)$,

$$\mathbb{E}[(\Psi^{t+1} - \Psi^t)] \le \frac{\eta}{2\alpha_\Phi} \mathbb{E}[\|\boldsymbol{g}^t\|_*^2] - \langle \nabla F^t(\boldsymbol{x}^t), \boldsymbol{x}^* - \boldsymbol{x}^t \rangle. \tag{4}$$

Using the bound of the potential change due to Inequality (4) and linearity of expectation, we get

$$\mathbb{E}\left[\sum_{t=1}^{T}\big(\lambda F^t(\boldsymbol{x}^*) - \mu F^t(\boldsymbol{x}^t)\big)\right] \leq \Psi^1 + \sum_{t=1}^{T}\mathbb{E}\left[\lambda F^t(\boldsymbol{x}^*) - \mu F^t(\boldsymbol{x}^t) + \Psi^{t+1} - \Psi^t\right]$$

$$\leq \Psi^1 + \sum_{t=1}^{T}\mathbb{E}\Bigg[\underbrace{\lambda F^t(\boldsymbol{x}^*) - \mu F^t(\boldsymbol{x}^t) - \langle \nabla F^t(\boldsymbol{x}^t), \boldsymbol{x}^* - \boldsymbol{x}^t\rangle}_{\leq\, 0 \text{ since } F^t \text{ is } (\lambda,\mu)\text{-concave}} + \frac{\eta}{2\alpha_\Phi}\|\boldsymbol{g}^t\|_*^2\Bigg]$$

$$\leq \frac{1}{\eta}D_\Phi(\boldsymbol{x}^*\|\boldsymbol{x}^1) + \frac{\eta}{2\alpha_\Phi}\sum_{t=1}^{T}\mathbb{E}[\|\boldsymbol{g}^t\|_*^2]. \tag{5}$$

If the norms $\|\boldsymbol{g}^t\|_*$ are bounded by $L_g$ and $D_\Phi(\boldsymbol{x}^*\|\boldsymbol{x}^1)$ is bounded by $G^2$ then

$$\mathbb{E}\left[\sum_{t=1}^{T}F^t(\boldsymbol{x}^t)\right] \geq \frac{\lambda}{\mu}\sum_{t=1}^{T}F^t(\boldsymbol{x}^*) - \frac{1}{\mu\cdot\eta}G^2 - \frac{\eta}{\mu\cdot 2\alpha_\Phi}TL_g^2$$

Choosing $\eta = \frac{G}{L_g}\sqrt{\frac{2\alpha_\Phi}{T}}$, the algorithm is $\left(\frac{\lambda}{\mu}, R(T)\right)$-regret where $R(T) = O\big(\frac{GL_g}{\mu}\sqrt{2\alpha_\Phi T}\big)$. $\qquad\square$

## 3.2 Bandit Algorithm

In this section, we consider the bandit setting in which at every time $t$ one can observe only the reward $f^t(\boldsymbol{x}^t)$ where $f^t$ is a bounded function defined on the convex set $\mathcal{K} = [0,1]^n$. W.l.o.g., assume that $f^t : [0,1]^n \to [0,1]$. In our algorithm, we will consider a discretization of $[0,1]^n$ and the multilinear relaxations of functions $f^t$ on these discrete points constructed as follows.

**Discretization and Multilinear Extension.** Let $f : [0,1]^n \to [0,1]$ be a function. Consider a lattice $\mathcal{L} = \{0, 2^{-M}, 2\cdot 2^{-M}, \ldots, \ell\cdot 2^{-M}, \ldots, 1\}^n$ where $0 \leq \ell \leq 2^M$ for some large parameter $M$ as a discretization of $[0,1]^n$. $M$ is a constant parameter to be chosen later. Note that each $x_i \in \{0, 2^{-M}, 2\cdot 2^{-M}, \ldots, \ell\cdot 2^{-M}, \ldots, 1\}$ can be uniquely decomposed as $x_i = \sum_{j=0}^{M} 2^{-j}y_{ij}$ where $y_{ij} \in \{0,1\}$. By this observation, we lift the set $[0,1]^n \cap \mathcal{L}$ to the $(n\times(M+1))$-dim space. Specifically, define a bijective *lifting* map $\mathtt{lift} : [0,1]^n \cap \mathcal{L} \to \{0,1\}^{n\times(M+1)}$ such that each point $(x_1, \ldots, x_n) \in \mathcal{K} \cap \mathcal{L}$ is mapped to the unique point $(y_{10}, \ldots, y_{1M}, \ldots, y_{n0}, \ldots, y_{nM}) \in \{0,1\}^{n\times(M+1)}$ where $x_i = \sum_{j=0}^{M} 2^{-j}y_{ij}$. Define function $\tilde{f} : \{0,1\}^{n\times(M+1)} \to [0,1]$ such that $\tilde{f}(\mathbf{1}_S) := f(\mathtt{lift}^{-1}(\mathbf{1}_S))$; in other words, $\tilde{f}(\mathbf{1}_S) = f(\boldsymbol{x})$ where $\boldsymbol{x} \in [0,1]^n \cap \mathcal{L}$ and $\mathbf{1}_S = \mathtt{lift}(\boldsymbol{x})$. Note that $\mathbf{1}_S$ with $S \subset [n\times(M+1)]$ is a $(n\times(M+1))$-dim vector with $(ij)^{th}$-coordinate equal to 1 if $(i,j) \in S$ and equal to 0 otherwise. Consider a multilinear extension $F : [0,1]^{n\times(M+1)} \to [0,1]$ of $\tilde{f}$ defined as follows.

$$F(\boldsymbol{z}) := \sum_{S\subset[n\times(M+1)]} \tilde{f}(\mathbf{1}_S) \prod_{(i,j)\in S} z_{ij} \prod_{(i,j)\notin S}(1 - z_{ij}).$$

By the definition, $F(\boldsymbol{z})$ can be seen as $\mathbb{E}[\tilde{f}(\mathbf{1}_S)]$ where the $(ij)^{th}$-coordinate of $\mathbf{1}_S$ equals 1 (i.e., $(\mathbf{1}_S)_{ij} = 1$) with probability $z_{ij}$.

**Algorithm description.** Our algorithm, formally given in Algorithm 1, is inspired by algorithm SCRIBLE [1] which has been derived in the context of bandit linear optimization. It has been observed that the gradient estimates of the functions in SCRIBLE are unbiased only if those functions are linear; and that represents a main obstacle in order to derive an algorithm with optimal regret guarantee $R(T) = \widetilde{O}(\sqrt{T})$. While aiming for the regret of $\widetilde{O}(\sqrt{T})$, in our algorithm, we overcome this obstacle by considering at every step the gradient estimate of the multilinear extension of the reward function (construction above). That gradient estimate will be indeed proved to be unbiased. Incorporating that gradient estimate to the scheme in [1] and following our approach, we prove the regret guarantee of the algorithm.

---
**Algorithm 1** Algorithm in the bandit setting.
---
1: Let $\Phi$ be a $\nu$-self-concordant function over $[0, 1]^{n \times (M+1)}$.
2: Let $\boldsymbol{z}^1 \in \text{int}([0, 1]^{n \times (M+1)})$ such that $\nabla \Phi(\boldsymbol{z}^1) = 0$.
3: **for** $t = 1$ to $T$ **do**
4:      Let $\boldsymbol{A}^t = [\nabla^2 \Phi(\boldsymbol{z}^t)]^{-1/2}$.
5:      Pick $\boldsymbol{u}^t \in \mathbb{S}_n$ uniformly random and set $\boldsymbol{y}^t = \boldsymbol{z}^t + \boldsymbol{A}^t \boldsymbol{u}^t$.
6:      Round $\boldsymbol{y}^t$ to a random point $\mathbf{1}_{S^t} \in \{0, 1\}^{n \times (M+1)}$ such that element $(i, j)$ appears in $S^t$ with probability $y_{ij}^t$.
7:      Play $\boldsymbol{x}^t = \texttt{lift}^{-1}(\mathbf{1}_{S^t})$ and receive the reward of $f^t(\boldsymbol{x}^t)$.
8:      Let $\boldsymbol{g}^t = -n(M+1)f^t(\boldsymbol{x}^t)(\boldsymbol{A}^t)^{-1}\boldsymbol{u}^t$ and compute the solution $\boldsymbol{z}^{t+1} \in [0, 1]^{n \times (M+1)}$ by applying the mirror descent framework on $F^t$ (Section 3.1). Specifically,

$$\boldsymbol{z}^{t+1} = \arg \max_{\boldsymbol{z} \in [0,1]^{n \times (M+1)}} \left\{ \langle \eta \boldsymbol{g}^t, \boldsymbol{z} - \boldsymbol{z}^t \rangle - D_\Phi(\boldsymbol{z} \| \boldsymbol{z}^t) \right\}.$$

---

**Analysis.** The remaining of the section is devoted to the analysis of Algorithm 1. For simplicity, until the end of this section, denote $m = n(M+1)$. Let $\mathbb{B}_m$ and $\mathbb{S}_m$ be the unit ball and the unit sphere in $m$ dimensions, respectively. For a constant $\delta$, define $\hat{F}_\delta(\boldsymbol{z}) := \mathbb{E}_{\boldsymbol{w} \in \mathbb{B}_m}[F(\boldsymbol{z} + \delta \boldsymbol{w})]$ where $\boldsymbol{w}$ is drawn from a uniform distribution over $\mathbb{B}_m$. We first prove some technical lemmas by exploiting properties of multilinear extensions. These lemmas are similar to those needed to prove the regret guarantee of SCRIBLE [1] but have some subtle differences because the functions we are considering are not linear (which is the case in [1]).

**Lemma 3** *It holds that $\hat{F}_\delta(\boldsymbol{z}) = F(\boldsymbol{z})$. Similarly, it also holds that $\mathbb{E}_{\boldsymbol{u} \in \mathbb{S}_m}[F(\boldsymbol{z} + \delta \boldsymbol{u})] = F(\boldsymbol{z})$.*

*Proof* Intuitively, the lemma holds since $F(\boldsymbol{z})$ is linear w.r.t $z_i$ for every $i$. In the following, we prove the first identity $\hat{F}_\delta(\boldsymbol{z}) = F(\boldsymbol{z})$. The second identity can be proved by exactly the same argument (using sphere instead of ball).

Consider a monomial $g_k(\boldsymbol{z}) = z_1 z_2 \ldots z_k$ for $1 \le k \le m$. We first prove by induction the claim that $\mathbb{E}_{\boldsymbol{w} \in \mathbb{B}_m(r)}[g_k(\boldsymbol{z} + \delta \boldsymbol{w})] = g_k(\boldsymbol{z})$ for every ball $\mathbb{B}_m(r)$ with radius $r$ in dimension $m$. (By this notation, $\mathbb{B}_m = \mathbb{B}_m(1)$.) The base case $k = 0$ is trivial. Assume that the induction hypothesis holds for $g_{k-1}(\boldsymbol{z})$. For any vector $\boldsymbol{w} \in \mathbb{B}_n(r)$, vector $\boldsymbol{w}' = (-v_k, \boldsymbol{w}_{-k})$ is also in $\mathbb{B}_n(r)$ and

$$g(\boldsymbol{z} + \delta \boldsymbol{w}) + g(\boldsymbol{z} + \delta \boldsymbol{w}') = (z_k + \delta v_k) \cdot g_{k-1}(\boldsymbol{z}_{-k} + \delta \boldsymbol{w}_{-k}) + (z_k - \delta v_k) \cdot g_{k-1}(\boldsymbol{z}_{-k} + \delta \boldsymbol{w}_{-k})$$
$$= 2z_k \cdot g_{k-1}(\boldsymbol{z}_{-k} + \delta \boldsymbol{w}_{-k})$$

Note that, for a given $|w_k|$, uniformly random vectors $(\pm w_k, \boldsymbol{w}_{-k})$ in the ball $\mathbb{B}_m(r)$ induce uniformly random vectors $\boldsymbol{w}_{-k}$ in the ball $\mathbb{B}_{m-1}(\sqrt{r^2 - |w_k|^2})$. Therefore,

$$\mathbb{E}_{\boldsymbol{w} \in \mathbb{B}_m(r)}[g_k(\boldsymbol{z} + \delta \boldsymbol{w})] = z_k \cdot \mathbb{E}_{v_k} \mathbb{E}_{\boldsymbol{w}_{-k} \in \mathbb{B}_{m-1}(\sqrt{r^2 - |v_k|^2})}[g_{k-1}(\boldsymbol{z}_{-k} + \delta \boldsymbol{w}_{-k})]$$
$$= z_k \cdot \mathbb{E}_{v_k}[g_{k-1}(\boldsymbol{z}_{-k})] = z_k \cdot g_{k-1}(\boldsymbol{z}_{-k}) = g_k(\boldsymbol{z})$$

where the second equality is due to the induction hypothesis. The claim follows.

As the multilinear extension is the sum of monomials multiplying with constant factors, the lemma holds because of the linearity of expectation. $\quad\square$

We restate here an useful lemma in [23].

**Lemma 4 ([23], Lemma 6.4)** *Let $\delta > 0$ be a fixed constant and $A \in \mathbb{R}^{m \times m}$ be an invertible matrix. Let $G(\boldsymbol{z}) := F(A\boldsymbol{z})$ and $\hat{G}_\delta(\boldsymbol{z}) := \mathbb{E}_{\boldsymbol{w} \in \mathbb{B}_m}[G(\boldsymbol{z} + \delta \boldsymbol{w})]$. Then, it holds that*

$$\mathbb{E}_{\boldsymbol{u} \in \mathbb{S}_m}\left[G(\boldsymbol{z} + \delta \boldsymbol{u})\boldsymbol{u}\right] = \frac{\delta}{m} \nabla \hat{G}_\delta(\boldsymbol{z})$$

*where the expectation is taken over uniform vector $\boldsymbol{u}$ in the $m$-dim unit sphere $\mathbb{S}_m$.*

**Lemma 5** *Let $A \in \mathbb{R}^{m \times m}$ be an invertible matrix. Define $\hat{F}(\boldsymbol{z}) := \mathbb{E}_{\boldsymbol{w} \in \mathbb{B}_m}\big[F(\boldsymbol{z} + A\boldsymbol{w})\big]$. Then it holds that*

*(i) $\hat{F}(\boldsymbol{z}) = F(\boldsymbol{z}) = \mathbb{E}_{\boldsymbol{u} \in \mathbb{S}_m}\big[F(\boldsymbol{z} + A\boldsymbol{u})\big]$.*

*(ii) $\nabla \hat{F}(\boldsymbol{z}) = m\mathbb{E}_{\boldsymbol{u} \in \mathbb{S}_m}\big[F(\boldsymbol{z} + A\boldsymbol{u})A^{-1}\boldsymbol{u}\big]$.*

*Proof* We prove the first part of the lemma. Again, we prove only the identity $F(\boldsymbol{z}) = \mathbb{E}_{\boldsymbol{w} \in \mathbb{B}_m}\big[F(\boldsymbol{z} + A\boldsymbol{w})\big]$; the identity $F(\boldsymbol{z}) = \mathbb{E}_{\boldsymbol{u} \in \mathbb{S}_m}\big[F(\boldsymbol{z} + A\boldsymbol{u})\big]$ can be proved using exactly the same arguments. Define $G(\boldsymbol{z}) := F(A\boldsymbol{z})$. Note that by this definition, $F(\boldsymbol{z} + A\boldsymbol{w}) = G(A^{-1}\boldsymbol{z} + \boldsymbol{w})$. The multilinear extension $F$ is the weighted sum of monomials, so is $G$. Therefore, by the same argument as in the proof of Lemma 3, for any $\boldsymbol{z}$ and $\delta$, it holds that $G(\boldsymbol{z}) = \mathbb{E}_{\boldsymbol{w} \in \mathbb{B}_m}\big[G(\boldsymbol{z} + \delta\boldsymbol{w})\big]$. Applying this identity with $\delta = 1$, we have

$$G(A^{-1}\boldsymbol{z}) = \mathbb{E}_{\boldsymbol{w} \in \mathbb{B}_m}\big[G(A^{-1}\boldsymbol{z} + \boldsymbol{w})\big] \qquad \Leftrightarrow \qquad F(\boldsymbol{z}) = \mathbb{E}_{\boldsymbol{w} \in \mathbb{B}_m}\big[F(\boldsymbol{z} + A\boldsymbol{w})\big] = \hat{F}(\boldsymbol{z}).$$

In the sequel, we prove the second part of the lemma. In fact, it can be proved using the same analysis in [23, Corollary 6.5]; we present it here for completeness. Define $\hat{G}(\boldsymbol{z}) := \mathbb{E}_{\boldsymbol{w} \in \mathbb{B}_m}\big[G(\boldsymbol{z} + \boldsymbol{w})\big]$. We have

$$m\mathbb{E}_{\boldsymbol{u} \in \mathbb{S}_m}\big[F(\boldsymbol{z} + A\boldsymbol{u})A^{-1}\boldsymbol{u}\big] = mA^{-1}\mathbb{E}_{\boldsymbol{u} \in \mathbb{S}_m}\big[F(\boldsymbol{z} + A\boldsymbol{u})\boldsymbol{u}\big] = mA^{-1}\mathbb{E}_{\boldsymbol{u} \in \mathbb{S}_m}\big[G(A^{-1}\boldsymbol{z} + \boldsymbol{u})\boldsymbol{u}\big]$$
$$= A^{-1}\nabla\hat{G}(A^{-1}\boldsymbol{z}) = A^{-1}A\nabla\hat{F}(\boldsymbol{z}) = \nabla\hat{F}(\boldsymbol{z})$$

where the third inequality is due to Lemma 4 with $\delta = 1$. $\qquad\square$

**Lemma 6** *If $f(\boldsymbol{x}) \leq 1$ for every $\boldsymbol{x} \in [0,1]^n$ then the corresponding multilinear extension $F$ is $2\sqrt{m}$-Lipschitz.*

*Proof* The proof comes directly from inspecting the derivatives. For each $1 \leq \ell \leq m$, we have

$$\left|\frac{\partial F(\boldsymbol{z})}{\partial z_\ell}\right| = \left|\sum_{S:\ell \in S} \tilde{f}(\mathbf{1}_S) \prod_{j \in S} z_j \prod_{j \notin S}(1 - z_j) - \sum_{S:\ell \notin S} \tilde{f}(\mathbf{1}_S) \prod_{j \in S} z_j \prod_{j \notin S}(1 - z_j)\right|$$
$$\leq \sum_{S:\ell \in S} \prod_{j \in S} z_j \prod_{j \notin S}(1 - z_j) + \sum_{S:\ell \notin S} \prod_{j \in S} z_j \prod_{j \notin S}(1 - z_j) \leq 2$$

where the first inequality is due to $\tilde{f}(\mathbf{1}_S) \leq 1$ for all $S \subset [m]$. Therefore, $\|\nabla F(\boldsymbol{z})\| \leq 2\sqrt{m}$. $\quad\square$

**Theorem 2** *Let $f^t : [0,1]^n \to [0,1]$ be the reward function at time $1 \leq t \leq T$ and let $F^t$ be the multilinear extension of the discretization of $f^t$ based on a lattice $\mathcal{L}$ (defined earlier). Assume that $F^t$'s are $(\lambda, \mu)$-concave and for every $\boldsymbol{x} \in [0,1]^n$, there exists $\overline{\boldsymbol{x}} \in \mathcal{L}$ such that $|f^t(\boldsymbol{x}) - f^t(\overline{\boldsymbol{x}})| \leq L \cdot 2^{-M}$ for every $1 \leq t \leq T$ (for example, $f^t$'s are L-Lipschitz). Then, by choosing $M = \log T$ and $\eta = O\big(\frac{1}{(nM)^{3/2} \cdot \sqrt{T}}\big)$ and $\Phi$ as a $O(nM)$-self-concordant function, Algorithm 1 achieves:*

$$\sum_{t=1}^{T} \mathbb{E}\big[f^t(\boldsymbol{x}^t)\big] \geq \frac{\lambda}{\mu} \cdot \max_{\boldsymbol{x} \in [0,1]^n} \sum_{t=1}^{T} f^t(\boldsymbol{x}) - O\big(\max\{\lambda/\mu, 1\}Ln^{3/2}(\log T)^{3/2}(\log \log T)\sqrt{T}\big).$$

*Proof* A crucial point in the analysis is the observation that the gradient estimator of the multilinear relaxation is unbiased. Specifically,

$$\mathbb{E}\big[\boldsymbol{g}^t\big] = \mathbb{E}_{\boldsymbol{u}^t}\mathbb{E}_{\boldsymbol{x}^t}\big[\boldsymbol{g}^t\big] = \mathbb{E}_{\boldsymbol{u}^t}\big[mF^t(\boldsymbol{y}^t)(\boldsymbol{A}^t)^{-1}\boldsymbol{u}^t\big] = \nabla\hat{F}^t(\boldsymbol{z}^t) = \nabla F^t(\boldsymbol{z}^t)$$

where the second equality holds since $\mathbb{E}\big[f^t(\boldsymbol{x}^t)\big] = F^t(\boldsymbol{y}^t)$ (by independent rounding and $F^t$ is multilinear relaxation of $f^t$); and the third and last equalities follow Lemma 5.

The remaining of the proof is similar to that in [23, Chapter 6] with some subtle differences because we consider the multilinear extensions of reward functions. Let $0 < \delta < 1/2$ be some small constant and consider the hypercube $[\delta, 1 - \delta]^m$. Note that, $[\delta, 1 - \delta]^m$ is convex and all balls of radius $\delta$ around points in $[\delta, 1 - \delta]^m$ are included in $[0,1]^m$.

Let $\boldsymbol{z}^*_\delta$ be the projection of $\boldsymbol{z}^*$ onto $[\delta, 1-\delta]^m$. Then by properties of projections, we have $\|\boldsymbol{z}^*_\delta - \boldsymbol{z}^*\| \leq \delta\sqrt{m}$. Moreover, as $|f^t(\boldsymbol{x})| \leq 1$ for every $\boldsymbol{x}, t$, and by definitions of the local norm, of $\boldsymbol{A}^t$ and $\boldsymbol{u}^t \in \mathbb{S}_m$, it holds that

$$\|\boldsymbol{g}^t\|^2_{*\boldsymbol{x}^t} \leq m^2 f^t(\boldsymbol{x}^t)^2 (\boldsymbol{u}^t)^\top ((\boldsymbol{A}^t)^{-1})^\top \nabla^{-2}\Phi(\boldsymbol{x}^t)(\boldsymbol{A}^t)^{-1}\boldsymbol{u}^t \leq m^2 \qquad (6)$$

Besides, by Lemma 6, the multilinear relaxations $F^t$'s are $(2\sqrt{m})$-Lipschitz for every $1 \leq t \leq T$.

For any $\boldsymbol{z}^* \in [0,1]^m$, we have

$$\frac{\lambda}{\mu}\sum_{t=1}^T F^t(\boldsymbol{z}^*) - \sum_{t=1}^T \mathbb{E}_{\boldsymbol{g}^t, \boldsymbol{u}^t}\big[F^t(\boldsymbol{y}^t)\big]$$

$$= \frac{\lambda}{\mu}\sum_{t=1}^T F^t(\boldsymbol{z}^*) - \sum_{t=1}^T \mathbb{E}_{\boldsymbol{g}^t}\big[F^t(\boldsymbol{z}^t)\big] \qquad \text{by Lemma } 5(i)$$

$$\leq \frac{\lambda}{\mu}\sum_{t=1}^T F^t(\boldsymbol{z}^*_\delta) - \sum_{t=1}^T \mathbb{E}_{\boldsymbol{g}^t}\big[F^t(\boldsymbol{z}^t)\big] + T(2\sqrt{m})\|\boldsymbol{z}^* - \boldsymbol{z}^*_\delta\| \qquad 2\sqrt{m}\text{-Lipschitz of } F^t$$

$$\leq \frac{1}{\mu \cdot \eta}D_\Phi(\boldsymbol{z}^*_\delta \| \boldsymbol{z}^1) + \frac{\eta}{\mu \cdot 2\alpha_\Phi}\sum_{t=1}^T \|\boldsymbol{g}^t\|^2_* + 2\delta m T \qquad \text{by Theorem } 1$$

$$= \frac{1}{\mu \cdot \eta}\big[\Phi(\boldsymbol{z}^*_\delta) - \Phi(\boldsymbol{z}^1) - \langle \nabla\Phi(\boldsymbol{z}^1), \boldsymbol{z}^*_\delta - \boldsymbol{z}^1 \rangle\big] + \frac{\eta}{\mu \cdot 2\alpha_\Phi}m^2 T + 2\delta m T$$

$$\leq \frac{1}{\mu \cdot \eta}\big[\Phi(\boldsymbol{z}^*_\delta) - \Phi(\boldsymbol{z}^1)\big] + \frac{\eta}{\mu \cdot 2\alpha_\Phi}m^2 T + 2\delta m T \qquad \text{since } \nabla\Phi(\boldsymbol{z}^1) = 0$$

$$\leq \frac{\nu}{\mu \cdot \eta}\log\frac{1}{1 - \pi_{\boldsymbol{z}^1}(\boldsymbol{z}^*_\delta)} + \frac{\eta}{\mu \cdot 2\alpha_\Phi}m^2 T + 2\delta m T \qquad \text{by Lemma } 2$$

$$\leq \frac{\nu \log\frac{1}{\delta}}{\mu \cdot \eta} + \frac{\eta}{\mu \cdot 2\alpha_\Phi}m^2 T + 2\delta m T \qquad \text{since } \boldsymbol{z}^*_\delta \in [\delta, 1-\delta]^m$$

$$= O\big(m\sqrt{\nu T}\log(mT)\big)$$

The second inequality holds since the unbiased stochastic gradients $\boldsymbol{g}^t$'s have corresponding dual local norm bounded by $m^2$. The last equality follows the choice $\eta = O(1/m\sqrt{\nu T}), \delta = O(1/m\sqrt{T})$.

Besides, by the property of multilinear extension and $\boldsymbol{x}^t$ is obtained from $\boldsymbol{y}^t$ by independently rounding, $F^t(\boldsymbol{y}^t) = \mathbb{E}_{\boldsymbol{x}^t}\big[f^t(\boldsymbol{x}^t)\big]$. Moreover, $F^t$ and $f^t$ have the same value on $\{0,1\}^n \cap \mathcal{L}$. Therefore,

$$\frac{\lambda}{\mu} \cdot \max_{\boldsymbol{x} \in [0,1]^n \cap \mathcal{L}}\sum_{t=1}^T f^t(\boldsymbol{x}) - \sum_{t=1}^T \mathbb{E}_{\boldsymbol{x}^t, \boldsymbol{u}^t, \boldsymbol{g}^t}\big[f^t(\boldsymbol{x}^t)\big] = \frac{\lambda}{\mu} \cdot \max_{\boldsymbol{z} \in [0,1]^m}\sum_{t=1}^T F^t(\boldsymbol{z}) - \sum_{t=1}^T \mathbb{E}_{\boldsymbol{u}^t}\big[F^t(\boldsymbol{y}^t)\big]$$

$$\leq O\big(m\sqrt{\nu T}\log(mT)\big) = O\big(m^{3/2}\sqrt{T}\log(mT)\big)$$

where the last equality is due to the fact that $O(m)$-self-concordant barrier exists [34, 9].

Let $\boldsymbol{x}^* \in \arg\min_{\boldsymbol{x} \in [0,1]^n}\sum_{t=1}^T f^t(\boldsymbol{x})$. By the theorem assumption, there exists $\overline{\boldsymbol{x}}^* \in [0,1]^n \cap \mathcal{L}$ such that $|f^t(\boldsymbol{x}) - f^t(\overline{\boldsymbol{x}})| \leq L \cdot 2^{-M}$. Hence, $\sum_{t=1}^T f^t(\boldsymbol{x}^*) \leq \sum_{t=1}^T f^t(\overline{\boldsymbol{x}}^*) + TL2^{-M}$. Choose $M = \log T$ (and recall that $m = n(M+1)$), we obtain the guarantee

$$\frac{\lambda}{\mu} \cdot \max_{\boldsymbol{x} \in [0,1]^n}\sum_{t=1}^T f^t(\boldsymbol{x}) - \sum_{t=1}^T \mathbb{E}_{\boldsymbol{x}^t, \boldsymbol{u}^t, \boldsymbol{g}^t}\big[f^t(\boldsymbol{x}^t)\big] \leq O\bigg(n^{3/2}M^{3/2}\sqrt{T}\log(nMT) + \frac{\lambda}{\mu}TL\frac{1}{2^M}\bigg)$$

$$= O\big(\max\{\lambda/\mu, 1\}Ln^{3/2}(\log T)^{3/2}(\log\log T)\sqrt{T}\big).$$

$\square$

## 4 Perturbed Discrete Time Fictitious Play in Smooth Auctions

We consider adaptive dynamics in auctions. In the setting, there is an underlying auction $\boldsymbol{o}$ and there are $n$ players, each player $i$ has a set of actions $\mathcal{A}_i$ (that can be arbitrarily large but finite) and

a valuation function $v_i$ taking values in $[0, 1]$. In each time step $1 \leq t \leq T$, each player $i$ selects a strategy which is a distribution in the space of distributions $\Delta(\mathcal{A}_i)$ according to some adaptive dynamic. The strategy profile at time $t$ is denoted as $\boldsymbol{\sigma}^t \in \Delta(\mathcal{A})$. Given the strategy profile $\boldsymbol{\sigma}^t$, the auction induces a social welfare $\mathrm{Sw}(\boldsymbol{o}, \boldsymbol{\sigma}^t) := \mathbb{E}_{\boldsymbol{a} \sim \boldsymbol{\sigma}^t}\left[\mathrm{Sw}(\boldsymbol{o}(\boldsymbol{a}); \boldsymbol{v})\right]$. In this setting, we study the performance of adaptive dynamics, especially the ones which are not guaranteed to fullfill the vanishing regret condition, and eventually design dynamics/auctions with performance guarantee. Among others, *fictitious play* is an interesting, widely-studied dynamic which attracts a lot of attention in the community. In this section, we will study the performance of a version of fictitious play in smooth auctions.

**Valuation-Oriented Fictitious Play.** Consider the Perturbed Discrete Time Fictitious Play (PDTFP) — a smooth version of Discrete Time Fictitious Play (for example, see [26]). Let $\Phi_i : \Delta(\mathcal{A}_i) \to \mathbb{R}$ for $1 \leq i \leq n$ be strongly convex functions ($\Phi_i$'s are not necessarily the same). Initially, each player chooses some arbitrary action. At time $t + 1$, given a strategy profile $\boldsymbol{\sigma}^t$ where $\sigma_i^t \in \Delta(\mathcal{A}_i)$ and perturbations $N_i^t : \Delta(\mathcal{A}_i) \to \mathbb{R}^+$ for $1 \leq i \leq n$ defined as $N_i^t(\sigma_i) = D_{\Phi_i}(\sigma_i \| \sigma_i^t)$, player $i$ selects a mixed strategy $\sigma_i^{t+1}$ such that

$$\sigma_i^{t+1} \in \arg \max_{\sigma_i \in \Delta(\mathcal{A}_i)} \mathbb{E}_{a_i \sim \sigma_i} \mathbb{E}_{\boldsymbol{a}_{-i}^t \sim \boldsymbol{\sigma}_{-i}^t}\left[v_i(\boldsymbol{a}_{-i}^t, a_i)\right] - \frac{1}{\eta} N_i^t(\sigma_i)$$

Equivalently,

$$\sigma_i^{t+1} \in \arg \max_{\sigma_i \in \Delta(\mathcal{A}_i)} \mathbb{E}_{a_i \sim \sigma_i} \mathbb{E}_{\boldsymbol{a}^t \sim \boldsymbol{\sigma}^t}\left[v_i(\boldsymbol{a}_{-i}^t, a_i) - v_i(\boldsymbol{a}^t)\right] - \frac{1}{\eta} N_i^t(\sigma_i), \tag{7}$$

since $\mathbb{E}\left[v_i(\boldsymbol{a}^t)\right]$ is already determined. One common example of perturbations is the *relative entropy* (or *Kullback-Leibler divergence*), defined as

$$N_i^t(\sigma_i) = \sum_{a \in \mathcal{A}_i} \sigma_i(a) \log \frac{\sigma_i(a)}{\sigma_i^t(a)}.$$

which is the Bregman divergence with the negative entropy function $\Phi_i(\sigma_i) = \sum_{a \in \mathcal{A}_i} \sigma_i(a) \log \sigma_i(a)$.

Let $V_i$ be the multilinear extension of the valuation $v_i$ of player $i$ (construction in Section 3.2) where now the corresponding lattice is the set of pure strategies $\mathcal{A}$. Note that the social welfare is the sum of all player valuations. Given an action profile $\boldsymbol{a}^t$, define $\nabla^t(\boldsymbol{a}^t) : \mathbb{R}^n \to \mathbb{R}$ such as

$$\langle \nabla^t(\boldsymbol{a}^t), \boldsymbol{x} \rangle = \sum_{i=1}^n \frac{\partial V_i(\boldsymbol{a})}{\partial a_i} \cdot x_i.$$

As $V_i$ is the multilinear extension of $v_i$, for every action $\boldsymbol{a}^*$ we have

$$\langle \nabla^t(\boldsymbol{a}^t), \boldsymbol{a}^* - \boldsymbol{a}^t \rangle = \sum_{i=1}^n \left[V_i(a_i^*, \boldsymbol{a}_{-i}^t) - V_i(a_i^t, \boldsymbol{a}_{-i}^t)\right].$$

The PDTFP dynamic can be cast as the mirror descent algorithm. By Equation (7) — the update rules of PDTFP dynamic — at every time step $t$, strategy profile $\boldsymbol{\sigma}^{t+1} = (\sigma_1^{t+1}, \ldots, \sigma_n^{t+1})$ is exactly the solution of the mirror descent update Equation (3):

$$\boldsymbol{\sigma}^{t+1} \in \arg \max_{\boldsymbol{\sigma} \in \Delta(\mathcal{A})} \mathbb{E}_{\boldsymbol{a} \sim \boldsymbol{\sigma}} \mathbb{E}_{\boldsymbol{a}^t \sim \boldsymbol{\sigma}^t}\left[\langle \nabla^t(\boldsymbol{a}^t), \boldsymbol{a} - \boldsymbol{a}^t \rangle\right] - \frac{1}{\eta} D_\Phi(\boldsymbol{\sigma} \| \boldsymbol{\sigma}^t)$$

where $\Phi$ is a strongly convex function such that $\Phi(\boldsymbol{\sigma}) = \sum_{i=1}^n \Phi_i(\sigma_i)$. Remark that, by the definition of $\Phi$, $D_\Phi(\boldsymbol{\sigma} \| \boldsymbol{\sigma}^t) = \sum_{i=1}^n D_{\Phi_i}(\sigma_i \| \sigma_i^t) = \sum_{i=1}^n N_i^t(\sigma_i)$. Again, if $\Phi_i$'s are negative entropy functions (so the perturbations $N_i^t$ are relative entropy functions) then $\Phi(\boldsymbol{\sigma}) = \sum_{i=1}^n \Phi_i(\sigma_i) = \sum_{i=1}^n \sum_{a \in \mathcal{A}_i} \sigma_i(a) \log \sigma_i(a)$. Note that the PDTFP dynamic associated to that choice of entropy function is usually called *smooth fictitious play* [19] (or *logit dynamic*).

**PDTFP dynamic in smooth auctions.** Given arbitrary perturbations $\Phi$, it is not clear whether the sequences of player actions in PDTFP dynamic are individually-vanishing-regret, i.e., satisfying condition (2). In particular, in the dynamic players are valuation-oriented whereas the condition (2) concerns player utilities. Hence, the welfare guarantee by [36, 40] for individually-vanishing-regret sequences cannot be applied. In the following, we show that although it is not known whether PDTFP dynamic are individually-vanishing-regret, they achieve a guarantee bound on the welfare in smooth auctions.

**Theorem 3** *If the underlying auction $o$ is a $(\lambda, \mu)$-smooth and $D_\Phi(\cdot \| \cdot)$ is bounded by $G^2$ then the PDTFP dynamic with parameter $\eta = O(G/\sqrt{T})$ achieves $\left(\frac{\lambda}{1+\mu}, R(T)\right)$-regret where $R(T) = O\left(\frac{G\sqrt{T}}{1+\mu}\right)$. In particular, if the perturbation is the relative entropy function then $R(T) = O\left(\frac{\sqrt{T \log(n|\mathcal{A}|)}}{1+\mu}\right)$.*

*Proof* The analysis follows closely the one of Theorem 1 with some modifications. For simplicity, without loss of generality, assume that the distributions $\overline{D}_1, \ldots, \overline{D}_n$ in the definition of smooth auctions (Definition 3) give rise to a pure strategy profile $\overline{a}$ and at any time step $t$, the PDTFP dynamic outputs a pure profile $a^t$. The analysis remains the same for general distributions/mixed profiles by putting additional expectations into some formula.

As the underlying auction is $(\lambda, \mu)$-smooth, given a fixed valuation profile $v$, there exists a strategy profile $\overline{a}$ such that for any profile $a$, it holds that

$$\sum_{i=1}^{n} u_i(\overline{a}_i, a_{-i}; v_i) \geq \lambda \cdot \text{OPT}(v) - \mu \cdot \text{REV}(a)$$

where $\text{OPT}(v)$ stands for the optimal welfare given the valuation profile $v$. We first derive an useful inequality based on the smoothness of the auction. We have

$$
\begin{aligned}
\langle \nabla^t(a^t), \overline{a} - a^t \rangle &= \sum_i \left[ V_i(\overline{a}_i, a^t_{-i}; v) - V_i(a_i, a^t_{-i}; v) \right] \\
&= \sum_i \left[ u_i(\overline{a}_i, a^t_{-i}; v_i) + p_i(\overline{a}_i, a^t_{-i}; v_i) \right] - \text{Sw}(a^t; v) \\
&\geq \lambda \cdot \text{OPT}(v) - \mu \cdot \text{REV}(a^t; v) - \text{Sw}(a^t; v) \\
&\geq \lambda \cdot \text{OPT}(v) - (1 + \mu) \cdot \text{Sw}(a^t).
\end{aligned}
\tag{8}
$$

The first inequality follows by the $(\lambda, \mu)$-smoothness and the non-negativity of payments $p_i$'s. The second inequality is obvious since the revenue is always smaller than the welfare. We remark that Inequality (8) is similar to (but not the same as) the notation of $(\cdot, \cdot)$-concavity since it can written as

$$\langle \nabla \text{Sw}(a^t), \overline{a} - a^t \rangle \geq \lambda \cdot \text{Sw}(a^*) - (1 + \mu) \cdot \text{Sw}(a^t)$$

where $a^*$ is the optimal strategy. Hence, there would be a connection between concavity and smoothness.

Define the potential as $\Psi^t = \frac{1}{\eta} D_\Phi(\overline{a} \| a^t)$. Note that here we use the Bregman divergence from the strategy $\overline{a}$ (induced by the smooth auction) to $a^t$ instead of the Bregman divergence from the optimal strategy $a^*$ to $a^t$ (as in Theorem 1). By the same arguments proving Inequality (4), we have

$$\eta\left(\Psi^{t+1} - \Psi^t\right) = D_\Phi(\overline{a} \| a^{t+1}) - D_\Phi(\overline{a} \| a^t) \leq -\eta \langle \nabla^t(a^t), \overline{a} - a^t \rangle + \frac{\eta^2}{2\alpha_\Phi} \|\nabla^t(a^t)\|_*^2$$

Given the valuation profile $\boldsymbol{v}$, let $\boldsymbol{a}^*$ be the action that gives the optimal welfare, i.e., $\textsc{Sw}(\boldsymbol{a}^*;\boldsymbol{v}) = \textsc{Opt}(\boldsymbol{v})$. Using the same arguments as in the proof of Theorem 1, we have

$$\sum_{t=1}^{T}\big(\lambda\textsc{Sw}(\boldsymbol{a}^*) - (1+\mu)\textsc{Sw}(\boldsymbol{a}^t)\big) \leq \Psi^1 + \sum_{t=1}^{T}\left[\lambda\textsc{Sw}(\boldsymbol{a}^*) - (1+\mu)\textsc{Sw}(\boldsymbol{a}^t) + \Psi^{t+1} - \Psi^t\right]$$

$$\leq \Psi^1 + \sum_{t=1}^{T}\left[\underbrace{\lambda\textsc{Opt}(\boldsymbol{v}) - (1+\mu)\textsc{Sw}(\boldsymbol{a}^t) - \langle\nabla^t(\boldsymbol{a}^t), \overline{\boldsymbol{a}} - \boldsymbol{a}^t\rangle}_{\leq 0 \text{ by Inequality (8)}} + \frac{\eta}{2\alpha_\Phi}\|\nabla_t(\boldsymbol{a}^t)\|_*^2\right]$$

$$\leq \frac{1}{\eta}D_\Phi(\overline{\boldsymbol{a}}\|\boldsymbol{a}^1) + \frac{\eta}{2\alpha_\Phi}\sum_{t=1}^{T}\|\nabla^t(\boldsymbol{a}^t)\|_*^2$$

Thus,

$$\sum_{t=1}^{T}\textsc{Sw}(\boldsymbol{a}^t) \geq \frac{\lambda}{1+\mu}\sum_{t=1}^{T}\textsc{Sw}(\boldsymbol{a}^*) - \frac{1}{(1+\mu)\eta}D_\Phi(\overline{\boldsymbol{a}}\|\boldsymbol{a}^1) - \frac{\eta}{(1+\mu)2\alpha_\Phi}\sum_{t=1}^{T}\|\nabla^t(\boldsymbol{a}^t)\|_*^2$$

Note that if player valuations are in the range $[0,1]$, then

$$\|\nabla^t(\boldsymbol{a}^t)\|_* \leq \|\nabla^t(\boldsymbol{a}^t)\|_\infty \leq 1.$$

By the theorem assumptions, $D_\Phi(\overline{\boldsymbol{a}}\|\boldsymbol{a}^1) \leq G^2$. Hence, choosing $\eta = O(G/\sqrt{T})$, the PDTFP dynamic achieves $\left(\frac{\lambda}{1+\mu}, R(T)\right)$-regret where $R(T) = O\big(\frac{G\sqrt{T}}{1+\mu}\big)$.

Consider the particular PDTFP dynamic with relative entropy perturbation. Function $\Phi(\boldsymbol{\sigma})$ is $\alpha_\Phi = \frac{1}{2\ln 2}$-strongly convex (due to Pinsker's inequality). Moreover, $D_\Phi(\overline{\boldsymbol{a}}\|\boldsymbol{a}^1) \leq \max_i \log(n|\mathcal{A}_i|) \leq \log(n|\mathcal{A}|)$. Therefore, choosing $\eta = O\big(1/\sqrt{T\log(n|\mathcal{A}|)}\big)$, the PDTFP dynamic with relative entropy perturbation achieves $\left(\frac{\lambda}{1+\mu}, R(T)\right)$-regret where $R(T) = O\big(\frac{\sqrt{T\log(n|\mathcal{A}|)}}{1+\mu}\big)$. $\qquad\square$

## 5 Online Simultaneous Second-Price Auctions with Reserve Prices

In this section, we are interested in the objective of maximizing the revenue. In the setting, there are $n$ bidders and $m$ items to be sold to these bidders. At each time step $t = 1, 2, \ldots, T$, the auctioneer selects reserve prices $r_i^t = (r_{i1}^t, \ldots, r_{im}^t)$ for each bidder $i$ where $r_{ij}$ is the reserve price of item $j$ for bidder $i$. Subsequently, every bidder $i$ picks a bid vector $b_i^t = (b_{i1}^t, \ldots, b_{im}^t)$ where $b_{ij}^t$ is the bid of bidder $i$ on item $1 \leq j \leq m$. Note that $b_{ij}^t$ and $b_{ij'}^t$ can be correlated. Then the auction for each item $1 \leq j \leq m$ works as follows: (1) remove all bidders $i$ with $b_{ij}^t < r_{ij}^t$; (2) run the second-price auction on the remaining bidders to determine the winner of item $j$; (3) charge the winner of item $j$ the larger of $r_{ij}^t$ and the second highest bid among the bids $b_{ij}^t$ of remaining bidders. Denote the revenue of selling item $j$ as $\textsc{Rev}_j(\boldsymbol{r}^t, \boldsymbol{b}^t)$ where $\boldsymbol{b}^t = (b_1^t, \ldots, b_n^t)$ and $\boldsymbol{r}^t = (r_1^t, \ldots, r_n^t)$. The revenue of the auctioneer at time step $t$ is $\textsc{Rev}(\boldsymbol{r}^t, \boldsymbol{b}^t) = \sum_{j=1}^{m}\textsc{Rev}_j(\boldsymbol{r}^t, \boldsymbol{b}^t)$. The goal of the auctioneer is to achieve the total revenue approximately close to that achieved by the best fixed reserve-price in hindsight $\sum_{j=1}^{m}\textsc{Rev}_j(\boldsymbol{r}^*, \boldsymbol{b}^t)$.

In the setting, by scaling, assume that all bids and reserve prices are in $\mathcal{K} = [0,1]^{n\times m}$. Consider the lattice $\mathcal{L} = \{\ell \cdot 2^{-M} : 0 \leq \ell \leq 2^M\}^{n\times m} \subset [0,1]^{n\times m}$ for some large parameter $M$ as a discretization of $[0,1]^{n\times m}$. Observe that for any reserve price vector $\boldsymbol{r}$, $|\textsc{Rev}(\boldsymbol{r}, \boldsymbol{b}) - \textsc{Rev}(\overline{\boldsymbol{r}}, \boldsymbol{b})| \leq m \cdot 2^{-M}$ where $\overline{\boldsymbol{r}}$ is a reserve price vector such that $\overline{r}_{ij}$ is the largest multiple of $2^{-M}$ smaller than $r_{ij}$ for every $i, j$ (for some large enough $M$). Therefore, one can approximate the revenue up to any arbitrary precision by restricting the reserve price on $\mathcal{L}$. We slightly abuse notation by denoting $\textsc{Rev}_j(\mathbf{1}_S, \boldsymbol{b})$ as $\textsc{Rev}_j(\boldsymbol{r}, \boldsymbol{b})$ where $\mathbf{1}_S = m(\boldsymbol{r})$ (recall that $m$ is the map defined in Section 3.2). Following Section 3.2, given a bid vector $\boldsymbol{b}$, the multilinear extension $\overline{\textsc{Rev}}$ of the revenue $\textsc{Rev}$ is defined as $\overline{\textsc{Rev}}(\cdot, \boldsymbol{b}) : [0,1]^{n\times m\times(M+1)} \to \mathbb{R}$ such that:

$$\overline{\textsc{Rev}}(\boldsymbol{z}, \boldsymbol{b}) = \sum_{S\subset[n\times m\times(M+1)]}\left(\sum_{j=1}^{m}\textsc{Rev}_j(\mathbf{1}_S, \boldsymbol{b})\right)\prod_{(i,j,k)\in S}z_{ijk}\prod_{(i,j,k)\notin S}(1 - z_{ijk}).$$

**Online bandit Reserve-Price Algorithm.** Initially, let $r^1$ be an arbitrary feasible reserve-price. At each time step $t \geq 1$,

(i) select $r^t$ or $\mathbf{0}$ each with probability 1/2 as the reserve-price;

(ii) receive the revenue corresponding to the selected reserve-price;

(iii) compute $r^{t+1}$ using Algorithm 1 with the following specification: in line 8 of Algorithm 1, replace $f^t(\boldsymbol{x}^t)$ by $2\mathrm{REV}(\boldsymbol{r}^t, \boldsymbol{b}^t)$ if the selected reserve-price is $\boldsymbol{r}^t$, or replace $f^t(\boldsymbol{x}^t)$ by 0 if the selected reserve-price is $\mathbf{0}$. (By doing that, the expected value of $\boldsymbol{g}^t$ in Algorithm 1 is $-\nabla \overline{\mathrm{REV}}(\boldsymbol{r}^t, \boldsymbol{b}^t)$.)

**Analysis.** In order to analyze the performance of this algorithm, we study the properties of some related functions and then derive the regret bound for the algorithm.

Fix a bid vector $\boldsymbol{b}$. Let $\boldsymbol{r}_j$ be a vector consisting of reserve prices on item $j$, i.e., $\boldsymbol{r}_j = (r_{1j}, \ldots, r_{nj})$. As $\boldsymbol{b}$ is fixed and the selling procedure of each item depends only on the reserve prices to the item, so for simplicity denote $\mathrm{REV}_j(\boldsymbol{r}, \boldsymbol{b})$ as $\mathrm{REV}_j(\boldsymbol{r}_j)$ and $\mathrm{REV}(\boldsymbol{r}, \boldsymbol{b})$ as $\mathrm{REV}(\boldsymbol{r})$. Define a function $h_j : \{0,1\}^{n \times (M+1)} \to \mathbb{R}$ such that $h_j(\mathbf{1}_T) = \max\{\mathrm{REV}_j(\mathbf{1}_T), \mathrm{REV}_j(\mathbf{1}_\emptyset)\} = \max\{\mathrm{REV}_j(\boldsymbol{r}), \mathrm{REV}_j(\mathbf{0})\}$ where $\boldsymbol{r}_j$ is the reserve price corresponding to $\mathbf{1}_T$ for $T \subset [n \times (M+1)]$. Let $H_j : [0,1]^{n \times (M+1)} \to \mathbb{R}$ be the multilinear extension of $h_j$. Moreover, define $H : [0,1]^{n \times m \times (M+1)} \to \mathbb{R}$ as the multilinear extension of $\max\{\mathrm{REV}(\boldsymbol{r}), \mathrm{REV}(\mathbf{0})\}$ defined as

$$H(\boldsymbol{z}) = \sum_{S \subset [n \times m \times (M+1)]} \max\{\mathrm{REV}(\mathbf{1}_S), \mathrm{REV}(\mathbf{1}_\emptyset)\} \prod_{(i,j,k) \in S} z_{ijk} \prod_{(i,j,k) \notin S} (1 - z_{ijk})$$

**Lemma 7** *It holds that $H(\boldsymbol{z}) = \sum_{j=1}^m H_j(\boldsymbol{z}_j)$ where $\boldsymbol{z}_j$ is the restriction of $\boldsymbol{z}$ to the coordinate related to item $j$.*

*Proof* As items are sold separately,

$$H(\boldsymbol{z}) = \sum_{S \subset [n \times m \times (M+1)]} \left( \sum_{j=1}^m h_j(\mathbf{1}_A) \right) \prod_{(i,j,k) \in S} z_{ijk} \prod_{(i,j,k) \notin S} (1 - z_{ijk})$$

where $A \subset [n \times (M+1)]$ is the restriction of $S$ on coordinates related to item $j$. Therefore,

$$H(\boldsymbol{z}) = \sum_{j=1}^m \sum_{U \subset [n \times (m-1) \times M]} \underbrace{\left[ \left( \sum_{A \subset [n \times (M+1)]} h_j(\mathbf{1}_A) \right) \prod_{(i,k) \in A} z_{ijk} \prod_{(i,k) \notin A} (1 - z_{ijk}) \right]}_{\text{independent of } U \text{ since the allocation of } j \text{ depends only on bids to item } j.}$$

$$\cdot \prod_{(i,j',k) \in U} z_{ij'k} \prod_{(i,j',k) \notin U, j' \neq j} (1 - z_{ij'k})$$

$$= \sum_{j=1}^m \left[ \sum_{A \subset [n \times (M+1)]} h_j(\mathbf{1}_A) \prod_{(i,k) \in A} z_{ijk} \prod_{(i,k) \notin A} (1 - z_{ijk}) \right]$$

$$\cdot \underbrace{\sum_{U \subset [n \times (m-1) \times (M+1)]} \prod_{(i,j',k) \in U} z_{ij'k} \prod_{(i,j',k) \notin U, j' \neq j} (1 - z_{ij'k})}_{=1}$$

$$= \sum_{j=1}^m \left[ \sum_{A \subset [n \times (M+1)]} h_j(\mathbf{1}_A) \prod_{(i,k) \in A} z_{ijk} \prod_{(i,k) \notin A} (1 - z_{ijk}) \right] = \sum_{j=1}^m H_j(\boldsymbol{z}_j)$$

$\square$

We will prove that $H$ is $(1,1)$-concave. By Lemma 7, it is sufficient to prove that property for every function $H_j$.

**Lemma 8** *Function $H_j$ is $(1,1)$-concave.*

*Proof* We prove that the condition $(1)$ of the $(1,1)$-concavity holds for all points in the lattice. As the multilinear extension can be seen as the expectation over these points, the lemma will follow. Fix a bid profile $\boldsymbol{b}_j = (b_{1j}, \ldots, b_{nj})$. Without loss of generality, assume that $b_{1j} \geq b_{2j} \geq \ldots \geq b_{nj}$. Let $\boldsymbol{r}_j$ and $\boldsymbol{r}_j^*$ be two arbitrary reserve price vectors. We will show that

$$\sum_{i=1}^{n} \left[ \max\{\text{Rev}_j(\boldsymbol{r}_{-ij}, r_{ij}^*), \text{Rev}_j(\boldsymbol{0})\} - \max\{\text{Rev}_j(\boldsymbol{r}_j), \text{Rev}_j(\boldsymbol{0})\} \right]$$
$$\geq \max\{\text{Rev}_j(\boldsymbol{r}_j^*), \text{Rev}_j(\boldsymbol{0})\} - \max\{\text{Rev}_j(\boldsymbol{r}_j), \text{Rev}_j(\boldsymbol{0})\} \quad (9)$$

where $\boldsymbol{r}_{-ij}$ stands for the reserve price vectors on item $j$ without the reserve price of bidder $i$.

Observe that the revenue $\max\{\text{Rev}_j(\boldsymbol{r}_j'), \text{Rev}_j(\boldsymbol{0})\}$ for every reserve price $\boldsymbol{r}_j'$ is at least the second highest bid $b_{2j}$ (that is obtained in $\text{Rev}_j(\boldsymbol{0})$). Moreover, for any reserve price $\boldsymbol{r}_j'$ such that the auctioneer either (1) removes the first bidder (with highest bid) or (2) removes the second bidder and $r_{1j}' \leq b_{2j}$, the revenue

$$\max\{\text{Rev}_j(\boldsymbol{r}_j'), \text{Rev}_j(\boldsymbol{0})\} = \text{Rev}_j(\boldsymbol{0}).$$

Hence, $\max\{\text{Rev}_j(\boldsymbol{r}_j'), \text{Rev}_j(\boldsymbol{0})\} \neq \text{Rev}_j(\boldsymbol{0})$ if and only if $b_{2j} < r_{1j}' \leq b_{1j}$.

By these observations, we deduce that

$$\max\{\text{Rev}_j(\boldsymbol{r}_{-ij}, r_{ij}^*), \text{Rev}_j(\boldsymbol{0})\} \neq \max\{\text{Rev}_j(\boldsymbol{r}_j), \text{Rev}_j(\boldsymbol{0})\}$$

if and only if $i = 1$ and

- either $b_{2j} \leq r_{1j} \neq r_{1j}^* \leq b_{1j}$;
- or $r_{1j}^* \in (b_{2j}, b_{1j}]$ but $r_{1j} \notin (b_{2j}, b_{1j}]$;
- or inversely $r_{1j} \in (b_{2j}, b_{1j}]$ but $r_{1j}^* \notin (b_{2j}, b_{1j}]$.

Thus, proving Inequality $(9)$ is equivalent to showing that

$$\max\{\text{Rev}_j(\boldsymbol{r}_{-1j}, r_{1j}^*), \text{Rev}_j(\boldsymbol{0})\} - \max\{\text{Rev}_j(\boldsymbol{r}_j), \text{Rev}_j(\boldsymbol{0})\}$$
$$\geq \max\{\text{Rev}_j(\boldsymbol{r}_j^*), \text{Rev}_j(\boldsymbol{0})\} - \max\{\text{Rev}_j(\boldsymbol{r}_j), \text{Rev}_j(\boldsymbol{0})\}$$

**Case** 1: $b_{2j} \leq r_{1j} \neq r_{1j}^* \leq b_{1j}$. In this case, both sides are equal to $r_{1j}^* - r_{1j}$.

**Case** 2: $r_{1j}^* \in (b_{2j}, b_{1j}]$ but $r_{1j} \notin (b_{2j}, b_{1j}]$. In this case, both sides are equal to $r_{1j}^* - b_{2j}$

**Case** 3: $r_{1j} \in (b_{2j}, b_{1j}]$ but $r_{1j}^* \notin (b_{2j}, b_{1j}]$. In this case, both sides are equal to $b_{2j} - r_{1j}$

**Case** 4: the complementary of all previous cases. In this case, both sides are equal to 0.

Therefore, Inequality $(9)$ holds and so the lemma follows. $\square$

**Theorem 4** *The online bandit reserve price algorithm achieves the regret bound of* $\left(1/2, O\big(mn^{3/2}(\log T)^{3/2}(\log \log T)\sqrt{T}\big)\right)$.

*Proof* Consider an imaginary algorithm which is similar to our online reserve price algorithm but at every step $t$, its gain on item $j$ is $\max\{\text{Rev}_j(\boldsymbol{r}^t), \text{Rev}_j(\boldsymbol{0})\}$. (This algorithm is called imaginary since one cannot decide which reserve price between $\boldsymbol{r}^t$ and $\boldsymbol{0}$ is better when the bid vector is not known.) We verify the conditions of Theorem 2. The discretization satisfies the condition that for any given bids $\boldsymbol{b}$, for any reserve price $\boldsymbol{r}$, there exists a reserve price $\overline{\boldsymbol{r}}$ in the lattice which gives $|\text{Rev}(\boldsymbol{r}, \boldsymbol{b}) - \text{Rev}(\overline{\boldsymbol{r}}, \boldsymbol{b})| \leq m \cdot 2^{-M}$. Moreover, Lemma 8 shows the $(1,1)$-concavity of $H$. Therefore, applying Theorem 2, the imaginary algorithm achieves the regret bound of $(1, R(T))$ where

$$R(T) = O\big(mn^{3/2}(\log T)^{3/2}(\log \log T)\sqrt{T}\big).$$

As the online reserve price algorithm selects at every step $t$ either $\boldsymbol{r}^t$ or $\boldsymbol{0}$ with probability 1/2, the revenue of the algorithm is at least half that of the imaginary algorithm. The theorem follows. $\square$

# 6 Bandit No-Envy Learning in Auctions

In this section we consider the bandit item selection problem. In the problem, there are $m$ items and a player with monotone submodular valuation $v : 2^{[m]} \to [0, 1]$. At every time step $1 \leq t \leq T$, the player chooses a subset of items $S^t \subset [m]$ and the adversary picks adaptively (probably depending on the history up to time $t - 1$ but not on the current set $S^t$) a threshold vector $\boldsymbol{p}^t$. The player observes only the thresholds $p_j^t$ for $j \in S^t$ and gets reward $v(S^t) - \sum_{j \in S^t} p_j^t$. Without loss of generality, assume that $0 \leq v(S) \leq 1$ for all $S \subset [m]$ and also $0 \leq p_j^t \leq 1$ for all $t$ and $j$.

We seek an $r$-approximate no-envy learning algorithm for some constant $0 < r \leq 1$. That is, for any adaptively chosen sequence of threshold vectors $\boldsymbol{p}^t$ for $1 \leq t \leq T$, the sets $S^t$ for $1 \leq t \leq T$ chosen by the algorithm satisfy

$$\mathbb{E}\left[\sum_{t=1}^T \left(v(S^t) - \sum_{j \in S^t} p_j^t\right)\right] \geq \max_{S \subseteq [m]} \sum_{t=1}^T \left(r \cdot v(S) - \sum_{j \in S} p_j^t\right) - R(T)$$

where the regret $R(T) = o(T)$.

Let $V : [0, 1]^m \to \mathbb{R}^+$ be the multilinear relaxation of the monotone submodular valuation $v$. Formally,

$$V(\boldsymbol{z}) = \sum_{S \subset [m]} v(S) \prod_{j \in S} z_j \prod_{j \notin S} (1 - z_j).$$

It is well-known that if $v$ is monotone submodular then $V$ is also monotone and it satisfies the diminishing return property: $\nabla V(\boldsymbol{x}) \geq \nabla V(\boldsymbol{y})$ for all $\boldsymbol{x}, \boldsymbol{y} \in [0, 1]^m$ such that $\boldsymbol{x} \leq \boldsymbol{y}$. Here for two vectors $\boldsymbol{a}, \boldsymbol{b}$, we mean $\boldsymbol{a} \leq \boldsymbol{b}$ iff $a_j \leq b_j$ for all $j$.

The following lemma, which has been implicitly proved in [22], shows that the multilinear relaxation $V$ is $(1, 2)$-concave.

**Lemma 9 ([22])** *For every* $\boldsymbol{x}, \boldsymbol{y} \in [0, 1]^m$, *it holds that*

$$\langle \nabla V(\boldsymbol{x}), \boldsymbol{y} - \boldsymbol{x} \rangle \geq V(\boldsymbol{y}) - 2V(\boldsymbol{x})$$

*Proof* Given vector $\boldsymbol{x}, \boldsymbol{y}$, let $\boldsymbol{x} \vee \boldsymbol{y}$ be the vector such that its $i^{th}$ coordinate is $\max\{x_i, y_i\}$ for every $1 \leq i \leq n$. Similarly, let $\boldsymbol{x} \wedge \boldsymbol{y}$ be the vector such that its $i^{th}$ coordinate is $\min\{x_i, y_i\}$ for every $1 \leq i \leq n$.

For any vectors $\boldsymbol{x} \leq \boldsymbol{z}$, using the diminishing return property $\nabla V(\boldsymbol{x}) \geq \nabla V(\boldsymbol{x} + t(\boldsymbol{z} - \boldsymbol{x}))$ for $0 \leq t \leq 1$, we have

$$V(\boldsymbol{z}) - V(\boldsymbol{x}) = \int_0^1 \langle \boldsymbol{z} - \boldsymbol{x}, \nabla V(\boldsymbol{x} + t(\boldsymbol{z} - \boldsymbol{x})) \rangle dt$$

$$\leq \int_0^1 \langle \boldsymbol{z} - \boldsymbol{x}, \nabla V(\boldsymbol{x}) \rangle dt = \langle \boldsymbol{z} - \boldsymbol{x}, \nabla V(\boldsymbol{x}) \rangle$$

Therefore,

$$V(\boldsymbol{x} \vee \boldsymbol{y}) - V(\boldsymbol{x}) \leq \langle \boldsymbol{x} \vee \boldsymbol{y} - \boldsymbol{x}, \nabla V(\boldsymbol{x}) \rangle. \tag{10}$$

Similarly for vectors $\boldsymbol{x} \leq \boldsymbol{z}$, we have

$$V(\boldsymbol{z}) - V(\boldsymbol{x}) = \int_0^1 \langle \boldsymbol{z} - \boldsymbol{x}, \nabla V(\boldsymbol{x} + t(\boldsymbol{z} - \boldsymbol{x})) \rangle dt$$

$$\geq \int_0^1 \langle \boldsymbol{z} - \boldsymbol{x}, \nabla V(\boldsymbol{z}) \rangle dt = \langle \boldsymbol{z} - \boldsymbol{x}, \nabla V(\boldsymbol{z}) \rangle.$$

Therefore,

$$V(\boldsymbol{x} \wedge \boldsymbol{y}) - V(\boldsymbol{x}) \leq \langle \boldsymbol{x} \wedge \boldsymbol{y} - \boldsymbol{x}, \nabla V(\boldsymbol{x}) \rangle \tag{11}$$

Summing (10) and (11) and using the fact $(\boldsymbol{x} \vee \boldsymbol{y}) + (\boldsymbol{x} \wedge \boldsymbol{y}) = \boldsymbol{x} + \boldsymbol{y}$, we obtain

$$V(\boldsymbol{x} \vee \boldsymbol{y}) + V(\boldsymbol{x} \wedge \boldsymbol{y}) - 2V(\boldsymbol{x}) \leq \langle \boldsymbol{y} - \boldsymbol{x}, \nabla V(\boldsymbol{x}) \rangle.$$

As $V$ is monotone and non-negative, we deduce that

$$V(\boldsymbol{y}) - 2V(\boldsymbol{x}) \leq \langle \boldsymbol{y} - \boldsymbol{x}, \nabla V(\boldsymbol{x}) \rangle.$$

$\square$

In order to apply our framework, we first prove the guarantee of the mirror-descent algorithm similar to the one in Section 3.1.

**Mirror descent.** Let $\Phi$ be a $\alpha_\Phi$-strongly convex function w.r.t $\|\cdot\|$. Initially, let $\boldsymbol{z}^1$ is an arbitrary feasible point. At time step $t$, play $\boldsymbol{z}^t$ and receive the vector $\boldsymbol{p}^t$. Compute $-\boldsymbol{g}^t$ an unbiased estimate of $\frac{1}{2}\nabla\big(V(\boldsymbol{z}^t) - \langle \boldsymbol{p}^t, \boldsymbol{z}^t \rangle\big) = \frac{1}{2}\nabla V(\boldsymbol{z}^t) - \boldsymbol{p}^t$. Update $\boldsymbol{z}^{t+1}$ as follows:

$$\boldsymbol{z}^{t+1} = \arg \max_{\boldsymbol{z} \in [0,1]^m} \left\{ \langle \eta \boldsymbol{g}^t, \boldsymbol{z} - \boldsymbol{z}^t \rangle - D_\Phi(\boldsymbol{z} \| \boldsymbol{z}^t) \right\}$$

**Theorem 5** *Then the mirror descent algorithm above achieves*

$$\sum_{t=1}^T \left( V(\boldsymbol{z}^t) - \langle \boldsymbol{p}^t, \boldsymbol{z}^t \rangle \right) \geq \max_{\boldsymbol{z} \in [0,1]^n} \sum_{t=1}^T \left( \frac{1}{2} V(\boldsymbol{z}) - \langle \boldsymbol{p}^t, \boldsymbol{z} \rangle \right) - \frac{1}{\eta} D_\Phi(\boldsymbol{x}^* \| \boldsymbol{x}^1) - \frac{\eta}{2\alpha_\Phi} \sum_{t=1}^T \|\boldsymbol{g}^t\|_*^2$$

*Proof* The analysis in similar to that of Theorem 1. Let $\boldsymbol{z}^* \in \arg\max_{\boldsymbol{z} \in [0,1]^n} \sum_{t=1}^T \big(\frac{1}{2}V(\boldsymbol{z}) - \langle \boldsymbol{p}^t, \boldsymbol{z} \rangle\big)$. Define the potential as $\Psi^t = \frac{1}{\eta} D_\Phi(\boldsymbol{z}^* \| \boldsymbol{z}^t)$. By the same argument as in the analysis of Theorem 1, we have

$$D_\Phi(\boldsymbol{z}^* \| \boldsymbol{z}^{t+1}) - D_\Phi(\boldsymbol{z}^* \| \boldsymbol{z}^t) \leq \frac{\eta^2}{2\alpha_\Phi} \|\boldsymbol{g}^t\|_*^2 - \eta \langle \boldsymbol{g}^t, \boldsymbol{z}^* - \boldsymbol{z}^t \rangle \tag{12}$$

Using the bound of the potential change due to Inequality (12), we get

$$\sum_{t=1}^T \left( \frac{1}{2} V(\boldsymbol{z}^*) - \langle \boldsymbol{p}^t, \boldsymbol{z}^* \rangle - V(\boldsymbol{z}^t) + \langle \boldsymbol{p}^t, \boldsymbol{z}^t \rangle \right)$$

$$\leq \Psi_1 + \sum_{t=1}^T \left( \frac{1}{2} V(\boldsymbol{z}^*) - V(\boldsymbol{z}^t) - \langle \boldsymbol{p}^t, \boldsymbol{z}^* - \boldsymbol{z}^t \rangle + \Psi^{t+1} - \Psi^t \right)$$

$$\leq \Psi_1 + \sum_{t=1}^T \left( \frac{1}{2} V(\boldsymbol{z}^*) - V(\boldsymbol{z}^t) - \langle \boldsymbol{p}^t, \boldsymbol{z}^* - \boldsymbol{z}^t \rangle - \langle \boldsymbol{g}^t, \boldsymbol{z}^* - \boldsymbol{z}^t \rangle + \frac{\eta}{2\alpha_\Phi} \|\boldsymbol{g}^t\|_*^2 \right)$$

$$= \Psi_1 + \sum_{t=1}^T \left( \frac{1}{2} V(\boldsymbol{z}^*) - V(\boldsymbol{z}^t) - \frac{1}{2} \langle \nabla V(\boldsymbol{z}^t), \boldsymbol{z}^* - \boldsymbol{z}^t \rangle + \frac{\eta}{2\alpha_\Phi} \|\boldsymbol{g}^t\|_*^2 \right)$$

$$\leq \frac{1}{\eta} D_\Phi(\boldsymbol{x}^* \| \boldsymbol{x}^1) + \frac{\eta}{2\alpha_\Phi} \sum_{t=1}^T \|\boldsymbol{g}^t\|_*^2. \tag{13}$$

The third inequality holds because of the $(1,2)$-concavity of $V$, i.e., $V(\boldsymbol{z}^*) - 2V(\boldsymbol{z}^t) - \langle \nabla V(\boldsymbol{z}^t), \boldsymbol{z}^* - \boldsymbol{z}^t \rangle \leq 0$. The theorem follows. $\square$

Combining Theorem 5 and Theorem 2, we obtain the following result.

**Theorem 6** *Using Algorithm 1 with the specification that in line 8, replace $f^t(\boldsymbol{x}^t)$ by $\frac{1}{2}v(S^t) - \sum_{j \in S^t} p_j$, one gets a bandit algorithm with the following guarantee.*

$$\sum_{t=1}^T \left( v(S^t) - \sum_{j \in S^t} p_j^t \right) \geq \max_{S \subset [m]} \sum_{t=1}^T \left( \frac{1}{2} v(S) - \sum_{j \in S} p_j^t \right) - O\big(m^{3/2}(\log T)^{3/2}(\log\log T)\sqrt{T}\big).$$

# 7  Conclusion

In this paper, we have introduced a framework to design efficient online learning algorithms. Apart of standard regularity requirements (such as compact convex domain, Lipschitz, etc), a new crucial property is the $(\lambda, \mu)$-concavity. Designing efficient online learning algorithms is now reduced to constructing $(\lambda, \mu)$-concave offline algorithms (also with other standard regularity conditions). We show the applicability of the framework through applications in auction design. Due to the simplicity of the conditions, we hope that our approach would be useful in designing efficient online algorithms with approximate regret bounds for different problems.

## Footnotes

[1] A coverage function $v : 2^{[m]} \to \mathbb{R}^+$ has the form $v(S) = |\cup_{j \in S} A_j|$ where $A_1, \ldots, A_m$ are subsets of $[m]$.