[Reviews · NeurIPS 2020]

Review 1

Summary and Contributions: Please see the comments to authors section

Strengths: Please see the comments to authors section

Weaknesses: Please see the comments to authors section

Correctness: Please see the comments to authors section

Clarity: Please see the comments to authors section

Relation to Prior Work: Please see the comments to authors section

Reproducibility: Yes

Additional Feedback: The paper studies the problem of online convex optimization problem, except that the functions that arrive online are not really concave. They are "close to" concave, formalized by the paper as (lambda, mu) concavity. The idea is that in many problems of interest where the input functions are not concave, the paper discretizes the function and consider the multilinear extension of the discretized function, which happens to be (lambda, mu) concave for reasonable values of lambda and mu. The paper presents three applications to illustrate the value of their approach. The first of these is the analysis of adaptive dynamics on (lambda, mu) smooth games where previously high welfare was known to be guaranteed (i.e., average welfare of playing the dynamics over time is at least lambda/mu of the optimal welfare) only for dynamics that had vanishing regret for each player. The open question was whether this extended to other no-regret dynamics as well. The paper considers a particular version of fictitious play for which we don't know yet whether the individually vanishing regret condition holds, but the paper shows that the average welfare of playing that dynamics gets at least lambda/(1+mu) fraction of the optimal welfare. The second application is one where the goal is to maximize revenue through repeated personalized reserve pricing of m items to n players (so n*m reserve prices in each round). The auction being run is an eager second price auction with reserves for each item, and the auctioneer is able to observe only the total revenue he receives for a chosen vector of reserve prices. He aims to optimize his revenue. The paper shows how to obtain half of the optimal revenue. The third one is on envy free learning in auctions where the goal is to pick a set of items in each round and a reserve a reward of v(S_t) - \sum_{j \in S_t} p_j(t), where the prices p(t) are determined by an adversary. The goal is to be able to pick sets in sequence that get a good utility compared to the single best set in hindsight for general classes of valuations v. Earlier a 1-1/e approximation was known for coverage valuations. This paper shows how to get a 1/2 approximation for submodular valuations. The algorithm is inspired by the "Scribble" algorithm of Abernethy, Hazan and Rakhlin for bandit linear optimization, where the gradient estimation being unbiased was dependent on the functions being linear. This paper solves this issue by considering multilinear extensions of the functions whose gradient estimates are established to be unbiased. Overall this is a solid set of results, solving well-defined open questions. The extension of good average welfare results beyond no-regret dynamics is the most exciting direction, although it is not unclear how general the particular version of fictitious play considered in this paper is. [After author feedback]: Thanks for your response.


Review 2

Summary and Contributions: The paper presents a bandit algorithm which achieves a bi-criteria approximation guarantee; in particular, the reward (loss) of the algorithm is a constant approximation of the reward (loss) of the best fixed action in hindsight plus some additive error that is o(T). Importantly, this bandit algorithm holds promise to achieve the above guarantee in online learning settings with bandit feedback where the learner chooses actions from a continuous and high-dimensional space and when the reward (loss) function is not linear/concave (linear/convex). Indeed, it is shown that the algorithm can be applied in several auction settings involving multiple bidders and multiple items and improve somewhat substantially over existing guarantees. There is one definition and two algorithmic ingredients for the general algorithmic framework, which is then specialized for each auction application: - The definition: A function is called (lambda, mu)-concave if for every x and y: <grad F(x), y-x > >= lambda* F(y) - mu*F(x) Concave functions are (1,1)-concave, but increasing mu makes it easier to satisfy the inequality. A nice example shown in the paper is that the multilinear extension of a monotone submodular function is (1,2)-concave. - Algorithm 0: In an online optimization setting wherein the sequence of functions are (lambda,mu)-concave and Lipschitz and the learner receives stochastic gradient feedback, it is shown that online stochastic mirror descent achieves a bi-criterion regret guarantee with a multiplicative approximation factor of lambda/mu and and O( sqrt(T) ) additive error (where O( ) hides factors depending on the Lipschitzness, mu and the distance of the first choice of the algo to the best solution in hindsight). - Algorithm 1: Consider a bandit setting wherein the sequence of reward functions f_t : [0,1]^n -> [0,1] do not have special structure except that they are Lipschitz and that the multi-linear extension F_t of each f_t with respect to a discrete lattice over [0,1]^n (I will explain what these are shortly) is (lambda,mu)-concave. Then there is a poly-time bandit algorithm achieving a bi-criterion regret guarantee with with a multiplicative approximation factor of lambda/mu and O( sqrt(T) poly(log(t)) ) additive regret (where O() hides polynomial factors in n, Lipschitzness). The multi-linear extension of each f_t is defined as follows. Discretize [0,1]^n into cubelets whose edges are 2^{-M}. Each point in the lattice can be represented as a subset of {0,1}^{M+1}^n since every coordinate of a point in the lattice can be represented in binary as some element of {0,1]^{M+1}. Because the f_t's are Lipschitz restricting to the lattice is OK. Every f_t, thus can be replaced by some g_t: {0,1}^{M+1}^n -> [0,1]. Now the multilinear extension is defined in the standard way. For z in [0,1]^{M+1}^n set F_t(z) = E [g_t(x)], where every coordinate v of x is sampled independently according to the Bernoulli with bias z_v. Showing the guarantees for online stochastic mirror descent (Algorithm 0) is based on standard mirror descent analysis but the argument is incorrect; see below. I did not try to fix it. Constructing Algorithm 1 adapts SCRIBBLE of Abernethy et al COLT'08 from the linear to the multi-linear case. It is a zero-order method that obtains a stochastic estimate of the gradient of the function by querying it at a random point in an ellipsoid (constructed using a self-concordant function). The analysis follows standard arguments. The main contribution of the paper is proposing the algorithmic framework and showing that it can be used in a few auction settings to get improved guarantees. I would probably give the paper a solid accept. But there is a bug in the proof of the guarantees attained by online stochastic mirror descent. For now, I will give weak accept and I will update upwards or downwards depending on whether the authors correct the bug.

Strengths: The algorithmic framework to obtain bi-criterion regret guarantees in bandit settings where the action space is continuous and high-dimensional and the reward (loss) function is not linear/concave (linear/convex). Showing that the framework can be used in online learning settings motivated by auction design and improving somewhat substantially some existing guarantees.

Weaknesses: I think there is a bug in the proof of online stochastic mirror descent. On page 7 of the supplementary material, second inequality between lines 254 and 255. I don't think the inequality follows. Here is what I think is a counterexample (the argument would be OK if there was no projection): - take Phi to be 0.5*ell_2^2 - take all norms to be ell_2 - take theta_t=y_{t}=(0,1) - take theta_{t+1}=y_{t+1} = (0,0) - set K to be half plane where first coordinate is bigger than say 1 - thus x_t =(1,1) - the inequality does not hold, as you would need 1 >= 2

Correctness: There is a bug in the proof of the guarantees of online stochastic mirror descent. See above. The rest looks correct from where I inspected.

Clarity: yes

Relation to Prior Work: yes

Reproducibility: Yes

Additional Feedback: EDIT AFTER REBUTTAL: The proof is indeed correct. I was confused by the appearance of two thetas, written in slightly different ways, and carrying different meanings in the same equation. The inequality that was confusing me now appears to be trivially true. It might be nice to change the notation to avoid this possible confusion. In view of correctness, I am happy to raise my score to 7. The guarantees for (lambda, mu) concave functions are straightforward using properties of mirror descent, however their use in auction settings through gridding and multi-linear extensions is a good contribution that makes the paper a clear accept in my view.


Review 3

Summary and Contributions: The paper tackles non-stochastic bandit problems with a focus on auctions to which conventional online convex optimizations can not be applied, that is, the reward functions are non-convex. To this end, they consider lifting the search space and the reward functions to a higher dimension space by the multilinear extension. They introduce a new notion called (\lambda, \mu)-concavity and show that the mirror descent algorithm can achieve a \lambda/\mu-regret bound which is sublinear on T. Furthermore, they provide some applications of this framework including auctions.

Strengths: (\lambda,\mu)-concavity will be a strong tool for analyzing regret bounds because If only we know the parameters \lambda and \mu of the problems, we can achieve \lambda/\mu-regret by the mirror descent algorithm. Furthermore, even if reward functions do not satisfy (\lambda,\mu)-concavity, its multilinear extension may hold the concavity. Submodular functions are typical examples that its multilinear extension holds concavity, but does not hold it in itself, which implies the proposed framework can work in many applications such as auctions.

Weaknesses: If we do not know the parameters of \lambda and \mu, we can not bound the regret by the proposed framework. Thus, when we solve a problem using the framework, if we do not know the parameters, we need to prove it even if a reward function is a submodular function, except for some specific functions such as monotone submodular functions. Furthermore, a reward function does not necessarily hold concavity after applying multilinear extension.

Correctness: Due to the tight review schedule, I cannot check all proofs of the theorems and lemmas, but main theorems seems to be correct.

Clarity: The paper is well written. I enjoyed reading the manuscripts and learned a lot about around the work.

Relation to Prior Work: The paper clearly discusses about relations to previous works.

Reproducibility: Yes

Additional Feedback: I have some questions regarding weaknesses. 1. Does multilinear extension of any reward function hold (.,.)-concavity? It is well-known that the multilinear extension of any submodular function holds concavity, but I do not know about the other functions. If the above answer is no, is there any other function that holds concavity except for submodular functions? 2. Could you characterize \lambda and \mu by some parameters of submodular function? Even if a reward function holds submodularity, we need to prove the values of \lambda and \mu. This seems to be cumbersome. Thus, I wonder if we know some parameters of a submodular function, such as curvature, we can compute \lambda and \mu from it. 3. In section 4, Does Theorem 3 imply that if a smooth auction satisfies (\lambda,\mu)-smooth, then it also holds (\lambda,1+\mu)-concavity? PDTFP is described as an application of the proposed framework, but in the proof, just a similarity between concavity and smoothness is provided. Thus, I cannot understand that PDTFP is an application of concavity. If the answer is yes, adding a statement about it, such as a corollary, helps us the connection.


Review 4

Summary and Contributions: The paper considers a bandit (adversarial) setting where the learner selects an action (from [0,1]^n) at each time step t and obtains a reward depends on the selection of the adversary (reward functions) and the selected action. The paper aims to derive an online algorithm with sublinear regrets (w.r.t. overall time period T) for settings where the reward functions are (lambda, mu)-concave. Under such a setting, they show that (lambda/mu, O(sqrt(T)))-regret is obtainable based on the standard mirror descent algorithm and the idea of discretization + multilinear extensions. They also show that similar ideas and concavity assumptions can be used to provide regret for other applications related to auctions under different objectives (i.e., smooth auctions + welfare maximization, multi-dimension actions with reserve prices + revenue maximization, and non-envy learning in auctions + no-envy objective). For these settings, they obtain a reasonable approximation ratio and regret bound comparing to the existing results.

Strengths: I think the paper studies interest novel problems in bandits and auctions (even though the applicability of some settings in the paper is debatable). The proposed idea and techniques can be applied to broader future domains. Although I didn't verify every part of the proofs, I think the general approach is feasible and correct. The only mirror downside is the lack of lower bounds (which could be difficult to obtain but it could help the readers to determine the tightness of the bounds).  

Weaknesses: See above.

Correctness: See above.

Clarity: Yes. I think the paper is very well written (please fixed the typos and add in more definitions for the notations).

Relation to Prior Work: Yes. [Part 2 of the additional comments; part 1 is below] 2 Framework of Online Learning Could you add some text to discuss what you are trying to do below? 2.1 Regret of (λ, μ)-Concave Functions Please define your notations for the second sentence (alpha strong and norm 1) "let x1 is an arbitrary" -> fix "Let gt be an unbiased estimate " -> how do you obtain gt? and define the gradient part "standard mirror descent ..." -> please also define the product and KL After talking about MD, please talk about the theorem and its implications and consequences In Theorem 1: argmax add in the domain Please define all other notations, please How do you choose the initial point and the estimator? they seem to be pretty important in your bound 2.2 Bandit Algorithm "banding setting" -> ? "W.l.o.g." -> spell it out Can comment on the computability of F and the extension? It seems to be quite expensive Is the F^t (lambda, mu)-concave here? "self-concordant function" -> please define the meaning of such a function Would be good to mention MD in the algorithm description 3 Online Simultaneous Second-Price Auctions with Reserve Prices Before the analysis, bar REV is defined but then later on you are refining H(z) ... ignoring bar REV that you defined earlier; can you talk about the connection of H and bar REV at a higher level? Also think that, if it's possible, adding other applications (no-envy learning and fictitious play) in the main paper would be nice (and remove some of the proof in the last page of the paper) ******************************* After rebuttal **************************** We would like to thank the authors for providing the responses. I have read your responses. After taking your responses into consideration during our discussion, my opinion of the paper did not change. 

Reproducibility: Yes

Additional Feedback: [Part 1] Note: I am reviewing this top-down. Please look over my comments before addressing the reviews. Title Maybe add "Its Applications"? It sounds like it is two separate things right now? unless you want to say something else Abstract "online bandit learning" -> would add setting "the algorithm" -> where is the algorithm mentioned? maybe use a different word "polynomial-time" -> w.r.t to what parameters? " total gain" -> maybe use reward instead? "a new notion of concavity" -> in what context? what is a new notion being applied to? "performance guarantee" -> regret? "concavity parameters" -> what are them for instance? "updates follow " -> update of? "multilinear extensions of the reward functions" -> unclear to me how you can extend the reward functions "is nearly optimal" -> how do you know? do you have lower results to show this? "a version of fictitious play in smooth auctions" -> how do they connect to your bandit results? How are your results compare to the existing results in similar settings (if any)? 1 Introduction "Online Learning" -> lower case "dynamically evolving environments ..." -> good to add some examples (such as ...) "the hidden regularity ..." -> peculiar word choice "performance guarantee" -> w.r.t to which benchmark? "major research agenda in online learning" -> add in citations; may be good to say what and which directions; add in a sentence or two about recent applications "new regularity condition " -> which is? I would suggest phrasing your research direction motivated by some applications instead of motivated by the "regularity condition" ... as it currently read, it feels like you are looking for specific settings that fit the condition; I think a better motivation should be another way around "time step" -> finite or infinite horizon in your setting? "an algorithm chooses" -> it seems like the action space is continuous space of n dimensions? Is there a reason why your setting is restricted in the space of [0,1]? Would it be ok to say the reward instead of the function f^t? Would it be useful to provide some citations of the general online problems and the bandit assumption "algorithm is (r, R(T))-regret if" -> I think you should highlight x^t to be the policy of the algorithm; also what is R(T) in this case? maybe it would be good to rearrange them to reflect a more standard regret definition? Do you require your algorithm to hold for any T? "...notion of concavity" -> please highlight where it was used originally and its previous applications State and define gradient of F(x) and the operation in (1) "multilinear relaxation of a " -> what is the meaning of this? 1.1 The Main Algorithm "emphasis on auctions" -> it would be good to discuss the setting (if it's not defined later) "standard approaches have various limits" -> what are the approaches that have limits? there are bandit setting (e.g., adversarial) algorithms that do not assume such concavity (1) and (2) are typically setting in bandit theory "lifting the search space " -> search space for? "multilinear extensions" -> how do you perform this? "sufficiently dense lattice" -> what size is L? "hypercube in a high dimension space" -> how high is the dimension and how do you perform the mapping? I guess the hypercube should be large enough to accommodate all of the points I guess if you don't know f^t, how do you do the mapping precisely? "the previous ones" -> which one? what did they use? "classic mirror descent" -> add citations and descriptions "L-Lipschitz" -> define ... why is this an important and natural assumption for your problem? In Informal Theorem 1, would it be good to redefine your benchmark w.r.t to expectation earlier? Do you have any lower bound for this setting? 1.2 Applications to Auction Design "v_i" -> it is for the goods or items? "A_i" -> what are the actions? "an allocation " -> how many items? does it matter in our context? "quasi-linear utility model" -> here v_i is a function? "The social welfare of an auction" -> the social welfare is just the sum of the values What if we ignore the payment component? What is the bandit setting being played here? how does it connect to auctions? It is a bit unclear to me right now 1.2.1 Fictitious Play in Smooth Auctions "by normalization" -> is it a valid assumption? "selects a strategy" -> what is this strategy? define \delta(A_i) "to some given adaptive dynamic" -> why do they have play according to some dynamic? what if they don't? "given adaptive dynamic" -> I guess this is the pulling actions in bandit "of the algorithm and subsequently,"-> algorithm is the dynamic here? How natural is the setting here? This would require the players not to make individual actions but rather follow the dynamic of some algorithm; does it make sense for the individuals to play the game repeatedly? What about the selection of f^t? I am guessing each joint action is an action of the bandit setting, and the payoff is observed in the underlying games ... and you want to obtain some bounded regrets of the utilities "w.r.t" -> spell it out "several auctions in widely studied settings are smooth" -> which ones? In Definition 2, what is bar a on the lambda part? "smoothness framework" -> maybe add smoothness framework to definition 2 How hard is it to find a "individually-vanishing-regret sequence"? "fictitious play " -> good to say what it is "Perturbed Discrete Time Fictitious Play (PDTFP)" -> should state what it is "fraction of the optimal welfare" -> Informal Theorem 2 is saying something differently w.r.t. to the regret? unless I understood something; please explain Also, how does it connect to the actions being played at eq? It might not necessarily reach an eq. at the end of time T 1.2.2 Revenue maximization in Multi-Dimensional Environments "that achieved by the best fixed reserve-price auction" -> is this a realistic benchmark in auctions? "Follow-the-Perturbed-Leader strategy" -> describe; is this tight? I think this is a better motivation than the earlier example. I do question whether the repeated scenario is realistic. Do you have a lower bound for this result? Is this regret tight? When m = 1, yours is sqr(n) worst the existing results Also, the domain of the value is no longer between [0,1]; would be good to say something about it 1.2.3 Bandit No-Envy Learning in Auctions "notion of Walrasian equilibrium" -> provide citations "if no buyer envies" -> maybe a proper definition is needed "gets the reward of" -> even if the reward could be negative? is there an opt-out option? I found the objective a bit awkward -- how come r is only within v(S), it seems to be a bit more natural when r is outside of whole term instead of just v(S) ... do you have any result on that? Also, the domain of the value is no longer between [0,1]; would be good to say something about it "convex rounding scheme " -> which is? from LP-duality? Do you have any lower bound results? What about other classes of functions (e.g., additive and subadditive)? 1.3 Related Work "Online Learning, or Online Convex Optimization" -> lower case? For the previous applications in earlier subsections, I think it would be good to mention something about the new concavity definition and its connection to the applications [I ran out of space, see above for Part 1]

[Author Response · NeurIPS 2020]

We would like to thank all reviewers for insightful comments. In the response below, we address the main concern of
Reviewer 2. Due to the space limit, we cannot address other comments, which are indeed interesting and helpful. All
typos and suggestions related to the presentation and the writing of the paper will be fixed/included in the final version.

**Response to Reviewer 2.**   The main concern is whether there is a bug in the proof of the online stochastic mirror
descent, specifically the inequality between line 254 and line 255 in the supplementary material.
In fact, there is **no** bug in the proof.

Let's consider first the given example (copy below).

```
- take Phi to be 0.5*ell_2^2
- take all norms to be ell_2
- take theta_t=y_{t}=(0,1)
- take theta_{t+1}=y_{t+1} = (0,0)
- set K to be half plane where first coordinate is bigger than say 1
- thus x_t =(1,1)
- the inequality does not hold, as you would need 1 >= 2
```

By the choice of $\Phi(\cdot) = \frac{1}{2}\| \cdot \|_2^2$, we have $\nabla\Phi(\boldsymbol{x}^t) = \boldsymbol{x}^t$. By our notation, $\boldsymbol{\theta}^t = \nabla\Phi(\boldsymbol{x}^t)$, so $\boldsymbol{\theta}^t = \boldsymbol{x}^t$. Consequently,
we do not understand why in the example, $\boldsymbol{\theta}^t$ is taken to be $(0,1)$ and $\boldsymbol{x}^t$ can be $(1,1)$.

Besides, between line 254 and line 255 (in the supplementary material, i.e., full paper), $\boldsymbol{\theta}^{t+1}$ does not involve so we do
not understand the role of $\boldsymbol{\theta}^{t+1}$ in the example here. We try to guess whether the reviewer meant $\boldsymbol{\vartheta}^{t+1}$. However, even
with that guess, we do not see any contradiction.

In the following, we give the proof of the inequality between line 254 and 255 with very detail explanation. Recall that
$\Psi^t = \frac{1}{\eta}D_\Phi(\boldsymbol{x}^*\|\boldsymbol{x}^t)$. First, we observe that

$$\eta\left(\Psi^{t+1} - \Psi^t\right) = D_\Phi(\boldsymbol{x}^*\|\boldsymbol{x}^{t+1}) - D_\Phi(\boldsymbol{x}^*\|\boldsymbol{x}^t) \tag{1}$$

$$\leq D_\Phi(\boldsymbol{x}^*\|\boldsymbol{y}^{t+1}) - D_\Phi(\boldsymbol{x}^*\|\boldsymbol{x}^t) \tag{2}$$

$$= \Phi(\boldsymbol{x}^*) - \Phi(\boldsymbol{y}^{t+1}) - \langle\underbrace{\nabla\Phi(\boldsymbol{y}^{t+1})}_{\boldsymbol{\vartheta}^{t+1}}, \boldsymbol{x}^* - \boldsymbol{y}^{t+1}\rangle - \Phi(\boldsymbol{x}^*) + \Phi(\boldsymbol{x}^t) + \langle\underbrace{\nabla\Phi(\boldsymbol{x}^t)}_{\boldsymbol{\theta}^t}, \boldsymbol{x}^* - \boldsymbol{x}^t\rangle \tag{3}$$

$$= \Phi(\boldsymbol{x}^t) - \Phi(\boldsymbol{y}^{t+1}) - \langle\boldsymbol{\vartheta}^{t+1}, \boldsymbol{x}^t - \boldsymbol{y}^{t+1}\rangle - \langle\boldsymbol{\vartheta}^{t+1} - \boldsymbol{\theta}^t, \boldsymbol{x}^* - \boldsymbol{x}^t\rangle \tag{4}$$

$$= \Phi(\boldsymbol{x}^t) - \Phi(\boldsymbol{y}^{t+1}) - \langle\boldsymbol{\theta}^t, \boldsymbol{x}^t - \boldsymbol{y}^{t+1}\rangle + \langle\eta\boldsymbol{g}^t, \boldsymbol{x}^t - \boldsymbol{y}^{t+1}\rangle + \langle\eta\boldsymbol{g}^t, \boldsymbol{x}^* - \boldsymbol{x}^t\rangle \tag{5}$$

$$\leq -\frac{\alpha_\Phi}{2}\|\boldsymbol{y}^{t+1} - \boldsymbol{x}^t\|^2 + \eta\langle\boldsymbol{g}^t, \boldsymbol{x}^t - \boldsymbol{y}^{t+1}\rangle + \eta\langle\boldsymbol{g}^t, \boldsymbol{x}^* - \boldsymbol{x}^t\rangle \tag{6}$$

$$\leq \frac{\eta^2}{2\alpha_\Phi}\|\boldsymbol{g}^t\|_*^2 + \eta\langle\boldsymbol{g}^t, \boldsymbol{x}^* - \boldsymbol{x}^t\rangle \tag{7}$$

where

(1) by definition of $\Psi^t$;

(2) by the generalized Pythagorean property (Lemma 1);

(3) by the definition of the Bregman divergence;

(4) by notation $\boldsymbol{\vartheta}^{t+1} = \nabla\Phi(\boldsymbol{y}^{t+1})$ and $\boldsymbol{\theta}^t = \nabla\Phi(\boldsymbol{x}^t)$;

(5) using $\boldsymbol{\vartheta}^{t+1} = \boldsymbol{\theta}^t - \eta \cdot \boldsymbol{g}^t$ by the algorithm;

(6) using the $\alpha_\Phi$-strong convexity of $\Phi$, specifically, $\Phi(\boldsymbol{x}^t)-\Phi(\boldsymbol{y}^{t+1})-\langle\boldsymbol{\theta}^t, \boldsymbol{x}^t-\boldsymbol{y}^{t+1}\rangle \leq -\frac{\alpha_\Phi}{2}\|\boldsymbol{y}^{t+1}-\boldsymbol{x}^t\|^2$ since
$\Phi(\boldsymbol{y}^{t+1}) \geq \Phi(\boldsymbol{x}^t) + \langle\boldsymbol{\theta}^t, \boldsymbol{y}^{t+1} - \boldsymbol{x}^t\rangle + \frac{\alpha_\Phi}{2}\|\boldsymbol{y}^{t+1} - \boldsymbol{x}^t\|^2$ where recall $\boldsymbol{\theta}^t = \nabla\Phi(\boldsymbol{x}^t)$ and $-\langle\boldsymbol{\theta}^t, \boldsymbol{x}^t - \boldsymbol{y}^{t+1}\rangle =$
$\langle\boldsymbol{\theta}^t, \boldsymbol{y}^{t+1} - \boldsymbol{x}^t\rangle$;

(7) using Cauchy-Schwarz inequality $\langle\boldsymbol{a}, \boldsymbol{b}\rangle \leq \|\boldsymbol{b}\|\|\boldsymbol{a}\|_* \leq \|\boldsymbol{b}\|^2/2 + \|\boldsymbol{a}\|_*^2/2$, specifically $\langle\eta\boldsymbol{g}^t, (\boldsymbol{x}^t - \boldsymbol{y}^{t+1})\rangle \leq$
$\frac{\alpha_\Phi}{2}\|\boldsymbol{y}^{t+1} - \boldsymbol{x}^t\|^2 + \frac{\eta^2}{2\alpha_\Phi}\|\boldsymbol{g}^t\|_*^2$.

Remark that, as mentioned in the paper, our approach follows the potential argument of Bansal et Gupta [4], which has
been appeared recently in *Theory of Computing, pp. 1-32, vol 15, 2019*. In particular, the part related to the concern
is proved in their paper (page 19, paragraph "Potential change", `https://theoryofcomputing.org/articles/`
`v015a004/v015a004.pdf`). Note that they considered convex functions.

In conclusion, we believe that our proof is correct.

[Meta-Review · NeurIPS 2020]

This paper is interesting, at the junction of bandits and auctions. One reviewer had strong concerns, but they were mostly due to a (bad) choice of fonts and they were lifter after the rebuttal. Happy to suggest acceptance !